

# A new approach for simulating the paleo evolution of the Northern Hemisphere ice sheets

Rubén Banderas[1,2], Jorge Alvarez-Solas[1,2], Alexander Robinson[1,2], and Marisa Montoya[1,2]

[1]Universidad Complutense de Madrid (UCM)
[2]Instituto de Geociencias (UCM-CSIC)

*Correspondence to:* Rubén Banderas (banderas.ruben@fis.ucm.es)

**Abstract.**

The aim of this study is to assess and improve the methods currently used to force ice sheet models offline. To this end, three different synthetic transient forcing climatologies are developed for the past 120 kyr following a perturbative approach and applied to an ice-sheet model. The results are used to evaluate their consequences for simulating the paleo evolution of the Northern Hemisphere ice sheets. The first method follows the usual approach in which temperature anomalies relative to present are calculated by combining a present-day climatology with a simulated glacial-interglacial climatic anomaly field interpolated through an index derived from ice-core data. In the second approach the representation of millennial-scale climate variability is improved by incorporating a simulated stadial-interstadial anomaly field. The third is a refinement of the second one in which the amplitudes of both orbital and millennial-scale variations are corrected to provide a perfect agreement with a recent absolute temperature reconstruction over Greenland. The comparison of the three climate forcing methods highlights the tendency of the usual approach to overestimate the temperature variability over North America and Eurasia at millennial timescales. This leads to a relatively high Northern Hemisphere (NH) ice-volume variability on these timescales. Through enhanced ablation, this results in too low an ice volume throughout the last glacial period (LGP), below or at the lower end of the uncertainty range of estimations. Improving the representation of millennial-scale variability alone yields an important increase of ice volume in all NH ice sheets, but especially in the Fennoscandian ice sheet (FIS). Optimizing the amplitude of the temperature anomalies to match the Greenland reconstruction results in a further increase of the simulated ice-sheet volume throughout the LGP. Our new method provides a more realistic representation of orbital and millennial scale climate variability and represents an improvement in the transient forcing of ice sheets during the last glacial period. Interestingly, our new approach underestimates ice-volume variations on millennial timescales as indicated by sea-level records. This suggests that either the origin of the latter is not the NH or that processes not represented in our study, notably variations in oceanic conditions, need to be invoked to account for an important role of millennial-scale climate variability on millennial-scale ice-volume fluctuations. We finally provide here both our derived climate evolution of the LGP using the three methods as well as the resulting ice-sheet configurations. These could be of interest for future studies dealing with the atmospheric or/and oceanic consequences of transient ice-sheet evolution throughout the LGP, and as a source of climate input to other ice sheet models.



# 1   Introduction

The climate history of the late Quaternary is marked by alternating episodes of growth and decay of Northern Hemisphere (NH) ice sheets on orbital time scales (e.g. Hays et al., 1976; Imbrie et al., 1992), as evidenced by different proxy data. Geological and geomorphological data show that during the Last Glacial Period (LGP, ca. 110-10 ka BP) large fractions of North America and Eurasia were covered by ice sheets that reached their maximum extent and volume at the Last Glacial Maximum (LGM, ca. 21 ka BP; e.g. Clark and Mix, 2002; Dyke et al., 2002; Svendsen et al., 2004). Sea level reconstructions derived from coral dating (Bard et al., 1996) as well as from the isotopic signal recorded in marine sediments (Bond et al., 1993; Waelbroeck et al., 2002; Rohling et al., 2009; Grant et al., 2012) show substantial variations as a result of the waxing and waning of ice sheets, with differences relative to the present ranging between +6 m at the maximum of the Last Interglacial (ca. 125 ka BP) and -130 m at the LGM.

In addition to proxy data, glacial isostatic adjustment (GIA) models have been used to reconstruct the past temporal evolution of ice sheets (Peltier and Andrews, 1976). By inverting relative sea-level records and accounting for the isostatic deformation of the solid Earth in response to ice-mass changes and redistributions, these models have facilitated estimation of the global ice volume at the LGM (Yokoyama et al., 2000; Milne et al., 2002) and reconstruction of the sea-level equivalent (SLE) ice volume throughout different intervals around this period (Lambeck et al., 2000; Lambeck and Chappell, 2001; Lambeck et al., 2002, 2014). Recently they have been refined by applying additional constraints based on the available global positioning system (GPS) measurements of vertical motion of the Earth's crust. This technique has been used to simulate the spatial configuration of ice sheets during the last deglaciation (Peltier et al., 2015). However, GIA models fail to provide a unique solution for the temporal history of ice thickness.

Forward ice-sheet modelling can help overcome the intrinsic limitations of the GIA technique by directly simulating the paleo evolution of ice sheets. Ideally, Earth System Models (ESMs) including fully coupled ice-sheet components are the appropriate tools to simulate the past, as well as the present and future evolution of ice sheets. However, because of their high computational cost, the long-term simulation of ice sheets generally relies on simpler tools such as intermediate complexity climate models coupled to ice-sheet models (e.g. Bonelli et al., 2009; Deblonde and Peltier, 1991; Ganopolski and Calov, 2011; Marsiat, 1994; Peltier and Marshall, 1995).

An alternative and even simpler method is to use ice-sheet models forced offline by a time-varying climatology. These exercises are carried out on a regular basis, as they are needed to calibrate ice-sheet models, to assess model sensitivity to different parameters, and to compare the sensitivities of different models. To obtain adequate initial conditions for the ice sheet, a relatively long spin-up is required, involving one or more glacial cycles depending on the ice sheets involved. Because of the lack of continuous, accurate proxy data, a synthetic time-varying climatology is often built based on a combination of climate-model and proxy data or even based on simpler assumptions (Kleman et al., 2013) and used to force the ice-sheet model. Generally a perturbative approach is followed by combining the present-day climatology, obtained from observational data, with two extreme climate snapshots of the last glacial cycle, obtained from climate simulations for specific time slices, and a time index, derived from proxy records, often from the Greenland ice-core record (e.g., Dansgaard et al., 1993). The





time-varying climatology is obtained by adding the constant glacial-interglacial temperature anomalies, scaled with the time-dependent index, to the present-day climatology. A similar procedure is applied to precipitation but considering ratios rather than anomalies (e.g. Marshall et al., 2000, 2002; Charbit et al., 2002, 2007; Zweck and Huybrechts, 2005).

Zweck and Huybrechts (2005) suggested that until fully coupled, comprehensive ice-sheet and climate models are available,
the latter is probably the best method to simulate the long-term evolution of ice sheets. However, an important drawback of this approach is that it clearly misrepresents climate variability at millennial timescales. The reason for this is that the spatial glacial-interglacial anomaly field used is associated with orbital climatic variations, but scaled following the characteristic time evolution of the index. If based on the Greenland ice-core record, the latter includes not only orbital but also millennial-scale climate variability. The spatial patterns of these two modes of variability are clearly not the same, as indicated by a wealth of
models and data (see Section 3). As a result, the combination of a single glacial-interglacial temperature anomaly field with an ice-core derived index including both orbital and millennial scale variability can be expected to lead to a misrepresentation of climate variability and thus of the past evolution of NH ice sheets.

Here we illustrate the problems derived from this approach, and propose a new offline climate forcing method that attempts to better represent the characteristic pattern of millennial-scale climate variability. Ice core records (e.g. Dansgaard et al., 1993;
NGRIP members, 2004) as well as a wide range of coupled climate models (Ganopolski and Rahmstorf, 2001; Menviel et al., 2014; Peltier and Vettoretti, 2014; Banderas et al., 2015; Zhang et al., 2014, 2017) suggest that millennial scale variability during the LGP was associated with the transition between two different climatic regimes: a stadial and an interstadial state that differ in the location and/or strength of North Atlantic Deep Water (NADW) formation. Here we assume the stadial state represents the background glacial climate at the LGM, with NADW formation south of Iceland, and include the interstadial
state as an additional independent snapshot that represents a millennial-scale excitation away from the background state as a result of a northward shift and intensification of NADW formation. A synthetic time-varying temperature climatology is built by combining present-day observations, the simulated LGM anomalies relative to present, scaled by an orbital-timescale index, and the simulated stadial-interstadial anomalies, scaled by a millennial-timescale index. An important, model-dependent issue is the extent to which the orbital and millennial-scale anomaly fields are well captured, in particular their amplitudes. To account
for this, a refinement of the method is proposed consisting in a time-varying scaling of both temperature anomalies, orbital and millennial. We then compare the effect of the synthetic climatologies built through the three methods on the simulated evolution of NH ice sheets throughout the last glacial cycle.

The paper is organized as follows: in Section 2 the ice-sheet model and the three climate forcing methods used are described. In Section 3 the results of applying these methods to force the ice-sheet model are shown, and their capability to simulate the
evolution of the NH ice-sheets during the last glacial cycle is compared. Finally, the main conclusions are summarised in Section 4.



## 2   Methodology

### 2.1   The ice-sheet model description

The model used in this study is the GRISLI ice-sheet model, developed by Ritz et al. (2001). GRISLI has been used in a number of studies in different domains including Antarctica (Ritz et al., 2001; Philippon et al., 2006; Alvarez-Solas et al., 2011a), Greenland (Quiquet et al., 2012, 2013), and glacial NH ice sheets (Peyaud et al., 2007; Alvarez-Solas et al., 2011b, 2013). For this reason and because the focus of our study is the climate forcing used to drive the model, only a brief description is given here; further details about the model can be found in these previous studies.

GRISLI is a hybrid three-dimensional thermomechanical ice-sheet model combining the Shallow Ice Approximation (SIA, Hutter, 1983) for grounded ice and the Shallow Shelf Approximation (SSA, MacAyeal, 1989) for ice shelves and ice streams. It uses finite differences on a staggered Cartesian grid at a 40 km resolution, corresponding to 224×208 grid points for the NH domain, with 21 vertical levels. Initial topographic conditions are provided by surface and bedrock elevations built from the ETOPO1 dataset (Amante and Eakins, 2009) and ice thickness (Bamber et al., 2001). Surface boundary conditions include the surface mass balance (SMB) and basal melting. The SMB is given by the sum of accumulation and ablation, both of which are calculated from monthly surface air temperatures (SATs) and monthly total precipitation. As these variables are strongly influenced by topographic effects, GRISLI accounts for changes in elevation at each time step considering a linear atmospheric profile for temperature (Ohmura and Reeh, 1991) and an exponential dependency of precipitation on temperature. Accumulation is calculated by assuming that the fraction of solid precipitation is proportional to the fraction of the year with mean daily temperature below 2°C. The daily temperature is computed from monthly SATs assuming that the annual temperature cycle follows a cosine function. Ablation is calculated using the positive-degree-day (PDD) method (Reeh, 1989). Basal melting inland is determined through a recent reconstruction of the present-day geothermal heat flux (Shapiro and Ritzwoller, 2004), while in the ocean it is set to a fixed value of 2 m a$^{-1}$ in regions where depth is larger than 450 m and fixed to 0 m a$^{-1}$ in shallower areas to favour the growth of ice sheets during cold periods.

### 2.2   The forcing methods

Synthetic time-varying climatologies are built using three different methods. All three use a perturbative approach as explained above (Section 1) by combining the present-day (PD) climatology, obtained from observational data, with climate snapshots of the last glacial cycle, obtained from climate simulations for specific time slices, and a time index, derived from proxy records. In all cases the indices used were built based on two recent complementary temperature reconstructions over Greenland (Figure 1): one from the NGRIP ice-core record for the LGP (Kindler et al., 2014), and another one from several ice-core records for the Holocene (Vinther et al., 2009). Their combination (hereafter, the KV reconstruction) results in a continuous temperature reconstruction for Greenland for the past 120 kyr (Figure 1a). The present-day climatology (Figure 2a-c) is taken from the ERA-INTERIM reanalysis (Dee et al., 2011). The climatic snapshots are obtained from climate simulations with the CLIMBER-3α model (Montoya and Levermann, 2008; Banderas et al., 2015, see Sections 2.2.1-2.2.3); the resulting anomalies with respect to present (Figure 2d-i) have been corrected by elevation according to the ICE-5G topography (Peltier, 2004).





Oceanic temperatures are fixed in all experiments to present-day values to ensure that any ice sheet changes are exclusively due to the atmospheric forcing. Finally, sea-level variations are prescribed according to the reconstruction by Grant et al. (2012, Figure 1a). The specific details of each method are described below.

### 2.2.1 Method 1

The first method (hereafter M1) follows the usual approach used in many previous studies (Marshall et al., 2000, 2002; Charbit et al., 2002, 2007; Zweck and Huybrechts, 2005). The time-varying temperature and precipitation are given by

$$\boldsymbol{T}(t) \quad = \quad \boldsymbol{T}_0 + (1 - \gamma(t)) \cdot \boldsymbol{\Delta T}_{\mathrm{orb}} \tag{1}$$

$$\boldsymbol{P}(t) \quad = \quad \boldsymbol{P}_0 \cdot [\gamma(t) + (1 - \gamma(t)) \cdot \delta \boldsymbol{P}_{\mathrm{orb}}] \tag{2}$$

where $\boldsymbol{T}_0$ and $\boldsymbol{P}_0$ are the ERA-INTERIM present-day temperature and precipitation climatologies (Figure 2a-c), and $\boldsymbol{\Delta T}_{\mathrm{orb}} =$
$\boldsymbol{T}_{\mathrm{lgm}} - \boldsymbol{T}_{\mathrm{pd}}$ and $\delta \boldsymbol{P}_{\mathrm{orb}} = \boldsymbol{P}_{\mathrm{lgm}} / \boldsymbol{P}_{\mathrm{pd}}$ are the orbital temperature anomaly and precipitation ratio relative to the present day, respectively, obtained from previous equilibrium simulations for the preindustrial and LGM climates performed with the CLIMBER-3$\alpha$ model (Figure 2d-f, Montoya and Levermann, 2008). Bold symbols indicate two-dimensional spatial fields. $\gamma$ is the time index, based on the KV reconstruction, normalized between 0 and 1 for the LGM and the present-day, respectively (Figure 1a). Thus, the time index dictates both orbital and millennial-scale variability.

### 2.2.2 Method 2

The second method (M2) is similar to M1 but the temperature and precipitation variability are split into two spectral components, corresponding to orbital and millennial timescales, respectively. The time-varying climatology is now given by

$$\boldsymbol{T}(t) \quad = \quad \boldsymbol{T}_0 + (1 - \alpha(t)) \cdot \boldsymbol{\Delta T}_{\mathrm{orb}} + \beta(t) \cdot \boldsymbol{\Delta T}_{\mathrm{mil}} \tag{3}$$

$$\boldsymbol{P}(t) \quad = \quad \boldsymbol{P}_0 \cdot \{\alpha(t) + (1 - \alpha(t)) \cdot \delta \boldsymbol{P}_{\mathrm{orb}} \cdot [(1 - \beta(t)) + \beta(t) \cdot \delta \boldsymbol{P}_{\mathrm{mil}}]\} \tag{4}$$

Here $\boldsymbol{\Delta T}_{\mathrm{orb}}$ and $\delta \boldsymbol{P}_{\mathrm{orb}}$ are as in M1, and $\boldsymbol{\Delta T}_{\mathrm{mil}} = \boldsymbol{T}_{\mathrm{is}} - \boldsymbol{T}_{\mathrm{st}}$ and $\delta \boldsymbol{P}_{\mathrm{mil}} = \boldsymbol{P}_{\mathrm{is}} / \boldsymbol{P}_{\mathrm{st}}$ are the millennial temperature anomaly and precipitation ratio, respectively, for the interstadial relative to the stadial state. The stadial mode in our study is represented by the aforementioned LGM climate simulation with CLIMBER-3$\alpha$ (Montoya and Levermann, 2008), while the interstadial mode is taken from a recent glacial transient simulation performed with the same model under glacial climatic conditions, but with intensified NADW formation (Banderas et al., 2015). Finally, $\alpha$ and $\beta$ are two indices that separately modulate the
contribution of the orbital and millennial anomalies (Figure 1). $\alpha$ is obtained after applying a low-pass frequency filter ($f_c$ = 1/18 kyr$^{-1}$) based on a spectral decomposition to the original KV reconstruction and normalising the resulting signal to be consistent with the forcing equations (Eqs. 3 and 4); $\beta$ is obtained following a similar procedure but retaining the high frequency signal of the KV reconstruction.

Inspection of equations 1 and 3 shows that the difference between M1 and M2 is just

$$\beta(t) \cdot \boldsymbol{\Delta T}_{\mathrm{mil}} + \beta(t) \cdot \boldsymbol{\Delta T}_{\mathrm{orb}} = \beta(t) \cdot (\boldsymbol{T}_{\mathrm{is}} - \boldsymbol{T}_{\mathrm{pd}}) \tag{5}$$

that is, the difference between the interstadial and the present-day simulated fields, scaled by the millennial-scale $\beta$ index.




### 2.2.3 Method 3

M2 significantly underestimates the amplitudes of millennial-scale fluctuations at the NGRIP ice-core location, as compared to the KV reconstruction (see Figure 3 and section 3.1). This is a consequence of the attenuated magnitude of the orbital (LGM minus present-day) and, particularly, the millennial (interstadial minus stadial) temperature anomalies simulated by

the CLIMBER-$3\alpha$ model. To correct for this, method 3 (M3) introduces a refinement with respect to M2 that consists of an adjustment to the time-varying climatology in such a way that the resulting synthetic temperature time series at the NGRIP site exactly matches the KV reconstruction (Figure 3a). To this end, two additional amplification factors ($f_{\mathrm{orb}}$, $f_{\mathrm{mil}}$) are included in the equation that governs the temperature forcing (Eq. 6). Each factor is given by the ratio of the corresponding temperature anomaly component of the KV reconstruction (either orbital, $\Delta T_{\mathrm{orb}}^{KV}$, or millennial $\Delta T_{\mathrm{mil}}^{KV}$) to the corresponding temperature

anomaly component simulated by the climate model at the NGRIP location ($\Delta T_{\mathrm{orb}}(\mathrm{NGRIP})$, $\Delta T_{\mathrm{mil}}(\mathrm{NGRIP})$), respectively. We thus have:

$$\boldsymbol{T}(t) = \boldsymbol{T}_0 + (1 - \alpha(t)) \cdot \boldsymbol{\Delta T}_{\mathrm{orb}} \cdot f_{\mathrm{orb}} + \beta(t) \cdot \boldsymbol{\Delta T}_{\mathrm{mil}} \cdot f_{\mathrm{mil}} \tag{6}$$

where

$$f_{\mathrm{orb}} = \frac{\Delta T_{\mathrm{orb}}^{KV}}{\Delta T_{\mathrm{orb}}(\mathrm{NGRIP})} \tag{7}$$

and

$$f_{\mathrm{mil}} = \frac{\Delta T_{\mathrm{mil}}^{KV}}{\Delta T_{\mathrm{mil}}(\mathrm{NGRIP})} \tag{8}$$

Here, $\Delta T_{\mathrm{orb}}^{KV}$ represents the temperature difference between the PD and the LGM in the orbital component of the KV reconstruction whereas $\Delta T_{\mathrm{mil}}^{KV}$ is the maximum temperature amplitude of the millennial-scale component of the KV reconstruction.

$$\Delta T_{\mathrm{orb}}(\mathrm{NGRIP}) = T_{\mathrm{lgm}}(\mathrm{NGRIP}) - T_{\mathrm{pd}}(\mathrm{NGRIP}) \tag{9}$$

$$\Delta T_{\mathrm{mil}}(\mathrm{NGRIP}) = T_{\mathrm{is}}(\mathrm{NGRIP}) - T_{\mathrm{st}}(\mathrm{NGRIP}) \tag{10}$$

are, as in M2, the simulated orbital and millennial-scale temperature anomaly fields of Montoya and Levermann (2008) and Banderas et al. (2015), respectively, evaluated at the NGRIP ice-core location. In this way the KV reconstruction is exactly recovered at NGRIP (Figure 3). Clearly this introduces a scaling of the synthetic temperature amplitudes elsewhere too.

Finally, in order to keep the same structure as in the previous methods, the amplification factors are both included within the

so-called optimized indices ($\alpha^{\star}$, $\beta^{\star}$). Thus

$$\boldsymbol{T}(t) = \boldsymbol{T}_0 + (1 - \alpha^{\star}(t)) \cdot \boldsymbol{\Delta T}_{\mathrm{orb}} + \beta^{\star}(t) \cdot \boldsymbol{\Delta T}_{\mathrm{mil}} \tag{11}$$

$$\boldsymbol{P}(t) = \boldsymbol{P}_0 \cdot \{\alpha^{\star}(t) + (1 - \alpha^{\star}(t)) \cdot \delta \boldsymbol{P}_{\mathrm{orb}} \cdot [(1 - \beta^{\star}(t)) + \beta^{\star}(t) \cdot \delta \boldsymbol{P}_{\mathrm{mil}}]\} \tag{12}$$

with

$$\alpha^{\star}(t) = 1 - (1 - \alpha(t)) \cdot \frac{\Delta T_{\mathrm{orb}}^{KV}}{\Delta T_{\mathrm{orb}}(\mathrm{NGRIP})} \tag{13}$$

$$\beta^{\star}(t) = \beta(t) \cdot \frac{\Delta T_{\mathrm{mil}}^{KV}}{\Delta T_{\mathrm{mil}}(\mathrm{NGRIP})} \tag{14}$$





The amplification factors reflect the skill of the climate model to reproduce the characteristic spectral amplitudes of the KV reconstruction at the NGRIP site. Since the model tends to underestimate the KV reconstruction, $\alpha^\star$ and $\beta^\star$ are both found to increase the amplitudes of the orbital and millennial-scale fluctuations, respectively, relative to the original $\alpha$ and $\beta$ indices (Figure 1b, c).

## 3 Results

### 3.1 Reconstruction of the NH climate

To evaluate the capability of the different methods to provide a realistic forcing for the ice-sheet model, the resulting synthetic climatologies should be compared against reconstructions. However, continuous, high resolution NH temperature reconstructions spanning the entire last glacial cycle are scarce. Besides the Greenland ice-core record, they are limited to a few sea surface temperature (SST) reconstructions in the Mediterranean Sea. We now compare the performance of each method in these two regions. Then we discuss the specific features of each method in continental regions that are relevant for ice-sheet growth even though indirect measurements are not available.

As an initial proof of consistency, the synthetic temperature curves generated in the location of the NGRIP ice-core using each method are compared to the KV reconstruction (Figure 3a). M1 shows an almost perfect agreement with the KV reconstruction. This is due to the fact that the temperature evolution is dictated by $\gamma$ alone, which comes from the NGRIP record, and that the absolute amplitude, given by the LGM minus present temperature anomaly simulated by the CLIMBER-$3\alpha$ model, at the NGRIP location turns out to be very similar to the glacial-interglacial temperature amplitude ($\sim 15$ K) of the KV reconstruction (Eq. 1 and Figures 1 and 2).

In contrast, M2 strongly underestimates the amplitude of the KV reconstruction, particularly at millennial time scales. The reason for this is that the amplitude of stadial-interstadial temperature changes simulated by the CLIMBER-$3\alpha$ model at the NGRIP location ($\sim 7$ K) is smaller than those indicated by the KV reconstruction (up to 16.5 K). In the model actually the maximum temperature anomaly is placed over the Nordic seas, as opposed to off the southeast coast of Greenland, the location where glacial abrupt climate changes are thought to reach their maximum amplitude in terms of temperature (Voelker and Workshop Participants, 2002). Meanwhile, the exact agreement in the temperature evolution between M3 and the KV reconstruction is predetermined by construction (Section 2.2.3).

At lower latitudes, the climatologies obtained by the three methods are compared against the Western Mediterranean SST reconstruction of Martrat et al. (2004); Martrat (2007) (hereafter, the M2007 reconstruction; Figure 3b). This record was obtained from marine sediment core ODP 161-977A, located in the eastern basin of the Alboran Sea (see location in Figure 4a). It is a high-resolution, continuous record throughout the LGP showing the characteristic modulation of glacial-interglacial climate variability as well as the sequence of rapid temperature transitions corresponding to the Dansgaard-Oeschger (D/O) events in Greenland (Dansgaard et al., 1993). In the first part of the last glacial cycle all three methods are found to underestimate the M2007 record fluctuations, suggesting too low an amplitude of the glacial-interglacial temperature anomaly. As was found for the NGRIP location, M1 and M3 generally agree best with each other as well as with M2007, particularly within the second




half of the last glacial cycle (Figure 3b), where they both show a larger amplitude than the M2007 reconstruction. M2 generally tends to underestimate the amplitude of abrupt temperature transitions as recorded by M2007. In addition to M2007, additional SST reconstructions spanning the last glacial cycle are available for the Iberian Margin (Pailler and Bard, 2002; Salgueiro et al., 2010). In particular, similar results (not shown) are obtained from the comparison with the SST reconstruction in the

MD95-2040 core of Pailler and Bard (2002). Farther north, the SST reconstruction for sediment core MD95-2006 (Dickson et al., 2008) in the western margin of the British isles, although only covering part of the LGP, shares similarities with M1 and M3 (see Supplementary Material; Figure S1). Note that due to the higher heat capacity of the ocean compared to the atmosphere, SAT variations are bound to have a larger amplitude than SST variations, thus the comparison is not direct and an overestimation of the reconstructed SST by the synthetic SAT should be expected.

The lack of continuous reconstructions in NH continental areas hampers the evaluation of the temperature signal derived from the three methods. Nonetheless, the synthetic temperature timeseries obtained in two sites, in North America and Fennoscandia, respectively, are assessed (Figure 5). These sites correspond to areas covered by the Laurentide (LIS) and the Fennoscandian (FIS) ice sheets during the LGP, respectively (see locations in Figure 4a). Several aspects stand out that can be traced back to the structural differences among the methods. First, at orbital time scales, the temperature variations obtained by all methods in

both sites show warmer climate conditions at the Eemian with respect to the Holocene and colder temperatures during MIS 2 and MIS 4. By construction, M1 and M2 are identical at these time scales, while in M3 the orbital amplitude is larger, resulting in temperatures 2-5 K colder throughout most of the LGP. Second, at millennial time scales, the amplitudes of the temperature variations obtained with the three methods are very different in both locations. M1 and M2 show the largest and smallest amplitudes, respectively, with differences above 10 K in the most prominent transitions. As previously discussed, M1 and M2

differ only at the millennial scale, by an amount given by Eq. 5. Thus the difference between these two methods resides in the difference between the orbital and the millennial scale temperature anomaly fields used in M1 and M2, respectively, scaled by the $\beta$ index. This boils down to the difference between the present-day and the interstadial temperature fields used in M1 and M2, respectively. These generally result in much larger positive deviations in M1 that, as will be shown below, affect the ice growth. M3 shows variations with intermediate temperature amplitudes between M1 and M2, reflecting the fact that, even

with the refined scaling, the amplitude of the millennial temperature anomaly at these sites is much lower than the orbital one (Figure 2d, g).

  Finally, in M1 the amplitude of millennial scale fluctuations is very similar in both sites as a consequence of the nearly-symmetric temperature pattern around Greenland, with two centers of negative values of similar amplitude coinciding with the selected sites (Figure 2d). In contrast, in M2, and most notably in M3, the differences between the two sites are larger, with

larger amplitudes in the FIS than in the LIS site. This is a consequence of the more asymmetric millennial scale temperature anomaly, characterized by a single centre of positive values in the Nordic seas (Figure 2g).

## 3.2 Reconstruction of NH ice-sheets

The temporal evolutions of the simulated NH ice sheets that result from imposing the different methods in the GRISLI model all show the characteristic modulation by orbital climate variability over the last glacial cycle (Figure 6; Figures S2 and S3). Ice





volume increases from 120 ka BP throughout the LGP until around 20 ka BP, where it reaches its maximum value, subsequently decreasing throughout the Holocene until the present day.

Important differences are found among the three methods. For all ice sheets, M1 and M3 show the smallest and largest volumes throughout the LGP, respectively; M2 shows intermediate values between the two. As mentioned before, by construction, M1 and M2 are identical at orbital timescales, and differ at millennial timescales. The lower ice volume in M1 relative to M2 is due to the larger amplitude of its millennial-scale fluctuations, resulting from the large amplitude of its orbital spatial component. Although these sometimes lead to smaller temperatures with respect to the orbital background curve, in general they result in large positive anomalies that, through enhanced ablation, induce a disruption of the growth of large ice sheets in the NH. In contrast, at millennial timescales M2 shows a muted response of ice-volume variations in all ice sheets as a result of the small amplitude of its millennial-scale component. Finally, the higher volumes in M3 compared to M2 are a result of its larger orbital amplitude, that results in colder temperatures throughout most of the LGP in the NH (Figure 5), despite its larger millennial-scale temperature fluctuations. Differences between M1 and M3 reflect both the larger and smaller amplitude fluctuations at orbital and millennial timescales, respectively, in the latter case.

Throughout the LGP, differences in global SLE between the most extreme ice-volume cases, M1 and M3, are roughly constant around 20 m, and generally larger for the LIS than for the FIS. The intermediate case M2 follows more closely M1 in the LIS, but M3 in the FIS. Around 55 ka BP M1 shows a large ice-volume drop in the FIS that has no counterpart in the LIS (Figure 6c). M2, in contrast, shows a more gradual evolution. Since the difference between M1 and M2 is exclusively their millennial scale variability, this would suggest a more important role of their differential millennial scale variability in the FIS than in the LIS site. However, a simple explanation in terms of local temperature is not possible: at millennial timescales, the temperature difference between M1 and M2 (or M3) is actually smaller for the FIS than for the LIS (Figure 5). From 60-40 ka, the FIS ice volume shows a similar evolution in M1 and M3, with large sub-orbital ice-volume variability and decreasing trend compared to M2 that can be related to the strong millennial scale variability after D/O event number 14, around 60 ka. The large drop in the FIS ice volume in M1 at 55 ka BP appears to be linked to D/O event number 12, possibly that with the highest amplitude in the whole LGP. However this D/O event appears both in M1 and M3, and in the latter case it barely has an impact. Thus, a nonlinear response must be invoked to explain the larger impact of millennial-scale variability in M1 in the FIS. Since the magnitudes of the warmings at the LIS and the FIS sites in M1 associated to this D/O are very similar, one possibility is that the lower ice volume of the FIS in M1 around 40 ka leads to a larger reduction in response to the warming of this D/O event through positive feedbacks.

Finally, the deglaciation shows a different behaviour in the three methods. M1 shows a much more abrupt transition into the Holocene, with ice already vanishing by the beginning of this period. This is a consequence of the abrupt temperature evolution in NGRIP that, by construction, in M1 is extrapolated to the rest of the globe, leading to peak temperatures already reached at the beginning of the Holocene and subsequently decreasing. In contrast, M2 and M3 show a smoother temperature evolution at the NH ice-sheet sites (Figure 4) that also leads to a smoother deglaciation.

We now focus specifically on M3, which in our view provides the best time-varying climatology. The time slices of ice thickness and velocities simulated under M3 provide a consistent picture of the spatial structure of NH ice sheets throughout





the LGP (Figures 4 and 6). In particular, the present-day configuration is satisfactorily reconstructed, showing a unique ice sheet over Greenland with regions of intense ice flow predominantly distributed along its southeastern and the northwestern margins (Figure 4a). Full glacial climatic conditions lead to the growth of two additional vast masses of ice over North America and Eurasia (Figures 4c and d). On the one hand, the simulated North American ice sheet (NAIS) comprises a merged dome

that aggregates the LIS, the Innuitian (IIS) and the Cordilleran (CIS) divides in the western, northern and eastern parts of the continent, respectively. The spatial extent of the NAIS shows a good agreement with respect to that estimated in previous studies (e.g. Peltier et al., 2015). The complexity of the NAIS spatial configuration is also reflected in the map of simulated velocities that present two active ice streams in the vicinity of the Hudson Bay and in the area of the Gulf of St. Lawrence in accordance with recent reconstructions (Margold et al., 2015). Meanwhile, the FIS covers the entire Scandinavian region as

well as the British isles and a large fraction of the Barents and the Kara seas as suggested by geological an geomorphological constraints (Svendsen et al., 2004). During MIS3, the extension of the NAIS is reduced as compared to the LGM, with an ice-free corridor separating the LIS from the CIS (Figures 4e and f). The FIS exhibits a decline in terms of volume and extension, particularly in the southwestern sector of the FIS where the British isles and its surroundings alternate between glaciated and ice-free periods on millennial time scales as a result of glacial abrupt climate variability (see Supplementary Material).

## 15  4   Discussion and conclusions

In this study, two new methods to force ice-sheet models offline are presented and compared with the more traditional approach. Three different time-varying climatologies are developed for the past 120 kyr following a perturbative approach and applied to an ice-sheet model to evaluate their consequences for the paleo evolution of ice sheets. In the first case, following the usual approach, temperature anomalies relative to present are calculated by combining the present-day climatologies, a simulated

glacial-interglacial climatic anomaly field, and an index derived from ice-core data that includes orbital as well as millennial scale variability. In the second case, anomalies relative to present day are decomposed into an orbital and a millennial-scale component. Orbital variations are calculated as in the first case, but millennial-scale variations are calculated by including a simulated stadial-interstadial anomaly. The third case is a refinement of the second case in which the amplitudes of both orbital and millennial-scale variations are corrected to provide a perfect agreement with the Greenland ice-core record. We herein

focus essentially on the differences between the traditional and the novel, refined method.

The time series derived from these methods are compared at several locations with the available proxy data: the Greenland ice-core record and two SST reconstructions in the Mediterranean Sea. By construction, the new method provides a perfect agreement with the ice-core record, improving the performance of previous methods. The comparison with the Mediterranean SST reconstructions does not show a significant improvement. All methods produce comparable results over these sites, tending

to show larger amplitude fluctuations than the SST record, which is to be expected given the larger heat capacity of the ocean relative to the atmosphere.

However, the methods differ strongly in their performance in key areas for the development of NH ice sheets, as North America and Fennoscandia. In these areas traditional methods yield millennial scale fluctuations of very large amplitude,



comparable to those recorded in Greenland. Improving the representation of millennial-scale variability by including a stadial-interstadial anomaly field leads to a strong reduction in the amplitude of millennial scale temperature fluctuations by more than 10 K in the most prominent transitions. In addition, as a result of the scaling of the orbital temperature anomaly field, the amplitude of orbital variations is enhanced, leading to colder temperatures by about 5 K in most of the LGP. Finally,

the traditional method leads to a very similar amplitude of millennial scale fluctuations over the two main NH landmasses as a consequence of the nearly-symmetric temperature pattern around Greenland. In contrast, the improved millennial-scale temperature field leads to the emergence of differences between the temperature evolutions in these areas.

The lack of continuous reconstructions in NH continental areas precludes the evaluation of the temperature time series derived for these sites. However, the fact that in the traditional method the amplitude of temperature variations at sites such

as the LIS and the FIS is very similar to those of the Greenland ice-core record strongly suggests that these temperature fluctuations are overestimated. If the mechanism behind millennial-scale variability are transitions between states of reduced Atlantic meridional overturning circulation (AMOC), with southward shifted deep water formation (e.g. Alley et al., 1999; Böhm et al., 2015; Ganopolski and Rahmstorf, 2001; Henry et al., 2016; Sarnthein et al., 1994), it is difficult to conceive of a similar temperature amplitude in the centre of the LIS or the FIS as in Greenland. Proxy data actually suggest that Greenland is

the location where glacial abrupt climate changes reach their maximum amplitude in terms of temperature, decreasing farther south in the NH (Voelker and Workshop Participants, 2002). In contrast, the temperature fluctuations obtained in the new approach, with amplitudes of 30-50% of those of the Greenland ice-core record and larger values over the LIS, down and upstream of the North Atlantic, seem more realistic.

Our results show that the traditional method leads to the lowest ice volume values throughout the whole LGP. Indeed,

millennial-scale climate variability within the climate forcing enhances NH ice-volume variability on millennial timescales. This leads to an underestimation of ice volume throughout most of the LGP. Improving its representation alone (in M2) yields an important increase of ice volume in all NH ice sheets, but especially in the FIS. Additionally improving the orbital and millennial scale fields through the scaling is found to increase it further.

Although sea-level records provide essential information to interpret past ice-volume variations, continuous highly-resolved

sea-level reconstructions are scarce and frequently rely on an insufficient temporal control. In addition, they generally provide inferences of global sea-level changes. This complicates the evaluation of our simulated NH ice volume timeseries against the paleorecord. However, the contribution to sea level of individual ice sheets can be assessed at specific time slices such as the LGM, for which reconstructions are indeed available. Estimates of the SLE change at the LGM relative to present (see the reviews by Clark and Mix, 2002; Clark and Tarasov, 2014) range between 70 m (Tarasov et al., 2012) and 92 m

(Denton and Hughes, 1981) for the LIS and between 14 m (note this case is based on modelling, see Clark and Mix (2002) and references therein) and 34 m (Denton and Hughes, 1981) for the FIS. Thus the traditional method is well below the uncertainty range of ice-volume estimations for the LIS and its lower end for the FIS. In contrast, our new, refined method is closer to the uncertainty range for the LIS and well within it for the FIS. Interestingly, our new approach underestimates ice-volume variations on millennial timescales as indicated by sea-level records. This suggests that either the origin of the latter is not

the NH or that processes not represented in our study need to be invoked to account for an important role of millennial-scale





climate variability on millennial-scale ice-volume fluctuations. Variation in oceanic conditions, ignored in our study, are a likely candidate.

The climate model used to build the present-day, LGM, and interstadial fields used in this study is an intermediate complexity model (Montoya et al., 2005). Additional climate models could be used to test the validity of our results. Nevertheless, we do

not expect this to change our main conclusions. To the extent that orbital and millennial-scale anomaly fields are different, our new forcing method should provide a better representation of the climate of the LGP. We expect this result to be robust against the use of different climate models. The precise temperature and ice volume evolution could, nevertheless, be model dependent, and this is worth investigating with additional climate models, in particular more comprehensive ones. In the last years a rising number of state-of-the-art climate models have recently shown two different climatic regimes under glacial conditions (Peltier

and Vettoretti, 2014; Zhang et al., 2014, 2017). This study opens a new research pathway for these models which could take advantage of our new forcing method to investigate their skill to provide a synthetic reconstruction of the climate variability of the last glacial cycle and therefore investigate the evolution of NH ice sheets. One recommendation that emerges from our study is that, in case of unavailability of an interstadial simulated snapshot to force the ice-sheet model, the use of a low-pass filtered index from the ice-core record should provide a better forcing than the traditional method including the full variability.

Finally, the novelty of this work lies in the consideration of an additional climatic pattern associated with millennial-scale climate variability to reconstruct the climate variability of the last glacial-interglacial cycle for the whole NH. Our results reveal that an incorrect representation of the characteristic pattern of millennial-scale climate variability within the climate forcing not only affects NH ice-volume variations at millennial timescales, but has consequences for glacial-interglacial ice-volume changes too. Thereby our new forcing method contributes to clarify the still uncertain role of glacial abrupt climate change

in past ice volume variations, thus shedding light on the evolution of the NH ice sheets. As mentioned above, one aspect that remains to be assessed in this work is the role of the ocean; this should be in the scope of future work.

*Code and data availability.*    The code used to generate the synthetic climatologies of this study is based on the equations described within the manuscript. The specific scripts are available from the corresponding author upon request. The variables associated to the three synthetic time-varying climatologies originated in this study are available in this link: http://www.pik-potsdam.de/ robinson/ism-forcing. The evolution of

three representative glaciological variables has also been included in the repository as output netcdf files. Additional output variables related to our experiments can be requested from the corresponding author.

*Author contributions.*    R. Banderas carried out the simulations, analysed the results and wrote the manuscript. All other authors contributed to the design of the simulations, the analysis of the results and the writing of the manuscript.

*Competing interests.*    The authors declare that there are no competing interests related to this work.



*Acknowledgements.* This work was funded by the Spanish Ministerio de Economía y Competitividad through project MOCCA (Modelling Abrupt Climate Change, Grant CGL2014-59384-R). R. Banderas was funded by a PhD Thesis grant of the Universidad Complutense de Madrid. A. Robinson is funded by the Marie Curie Horizon2020 project CONCLIMA (Grant 703251). Part of the computations of this work were performed in EOLO, the HPC of Climate Change of the International Campus of Excellence of Moncloa, funded by MECD and

5    MICINN. This is a contribution to CEI Moncloa.



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



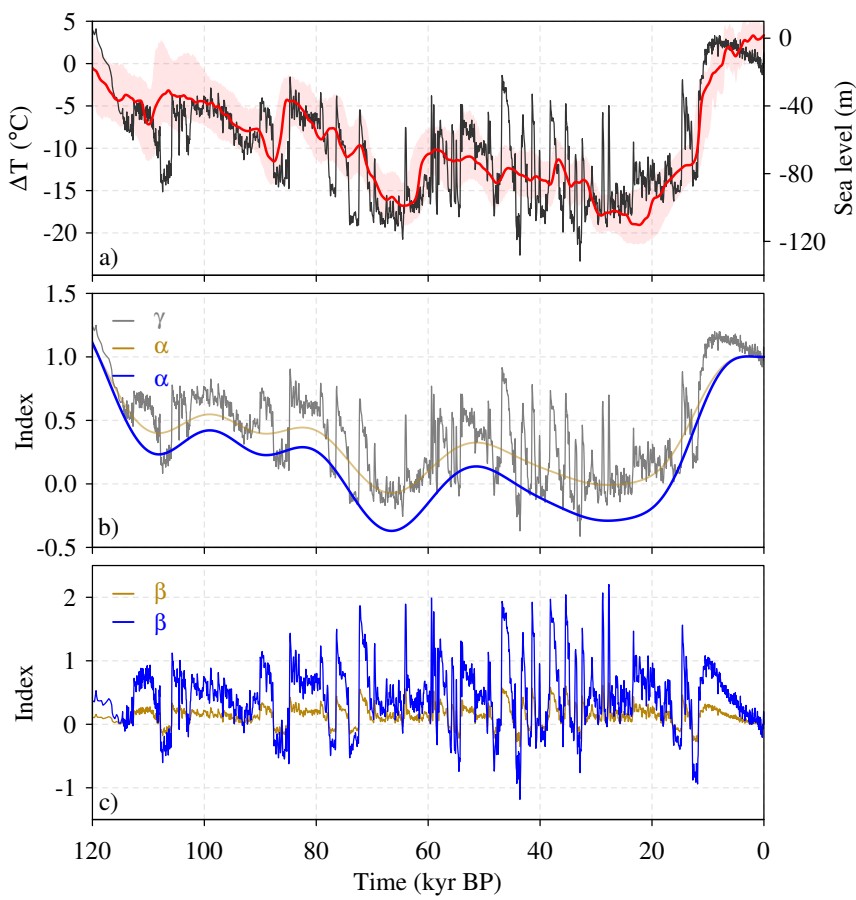

**Figure 1.** Temporal components of the three forcing methods: **a)** Sea level forcing (m) as estimated by Grant et al. (2012). The light red shaded area represents the 95% confidence level interval of the prescribed sea level reconstruction. The black curve shows the evolution of temperature anomalies (°C) relative to present at the NGRIP site over Greenland (75.1°N, 42.32°W) from which the index is derived (Vinther et al., 2009; Kindler et al., 2014); **b)** Index used in M1 ($\gamma$; gray) together with the orbital components of the indices used in M2 ($\alpha$; gold) and M3 ($\alpha^\star$; blue), respectively; **c)** Millennial components of the index used in M2 ($\beta$; gold) and M3 ($\beta^\star$; blue), respectively.





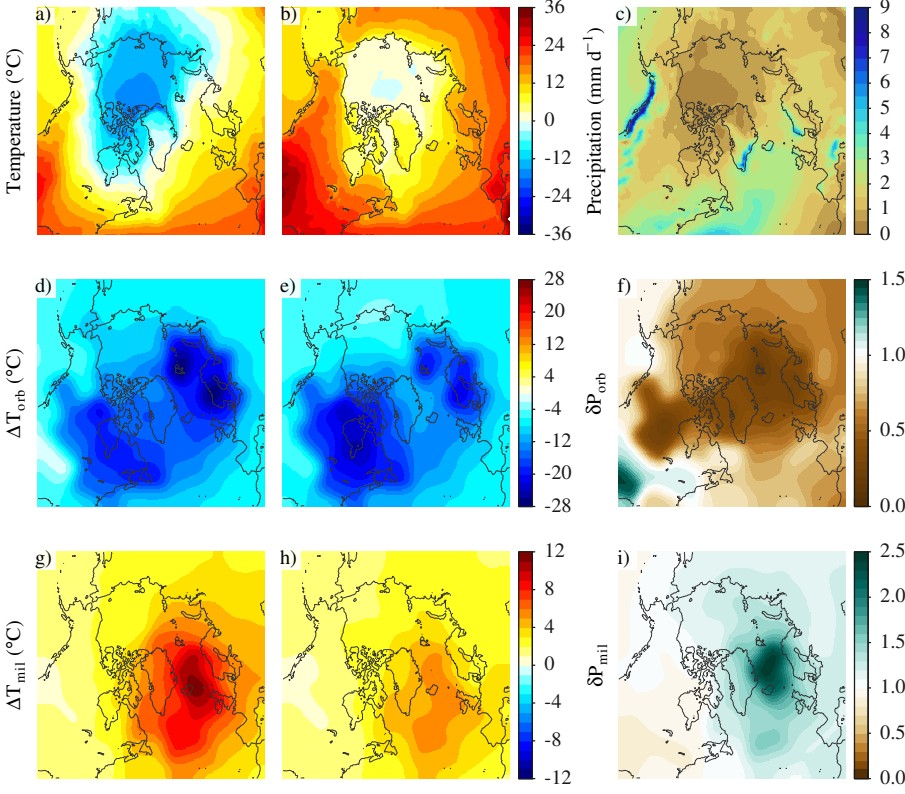

**Figure 2.** Spatial components of the different methods. The reference climate is based on the ERA-INTERIM (1981-2010) reanalysis (Dee et al., 2011) and consists of: **a)** annual SAT (°C); **b)** summer (JJA) SAT (°C) and **c)** annual precipitation (mm d$^{-1}$). The orbital component of the spatial forcing comprises the anomalies between the LGM and the present-day climates obtained from the CLIMBER-3$\alpha$ model (Montoya and Levermann, 2008): **d)** annual SAT (°C), **e)** summer (JJA) SAT (°C) and **f)** annual precipitation ratio ($\delta P_{orb} = P_{\mathrm{pd}}/P_{lgm}$). Panels **g)**, **h)** and **i)** show the same fields as in **d)**, **e)** and **f)** for the millennial component of the spatial forcing generated from the combination of the Is and the St climatic states simulated by CLIMBER3-$\alpha$ (Banderas et al., 2012, 2015). All variables have been corrected by elevation assuming a linear vertical atmospheric profile (see Section 2.1).





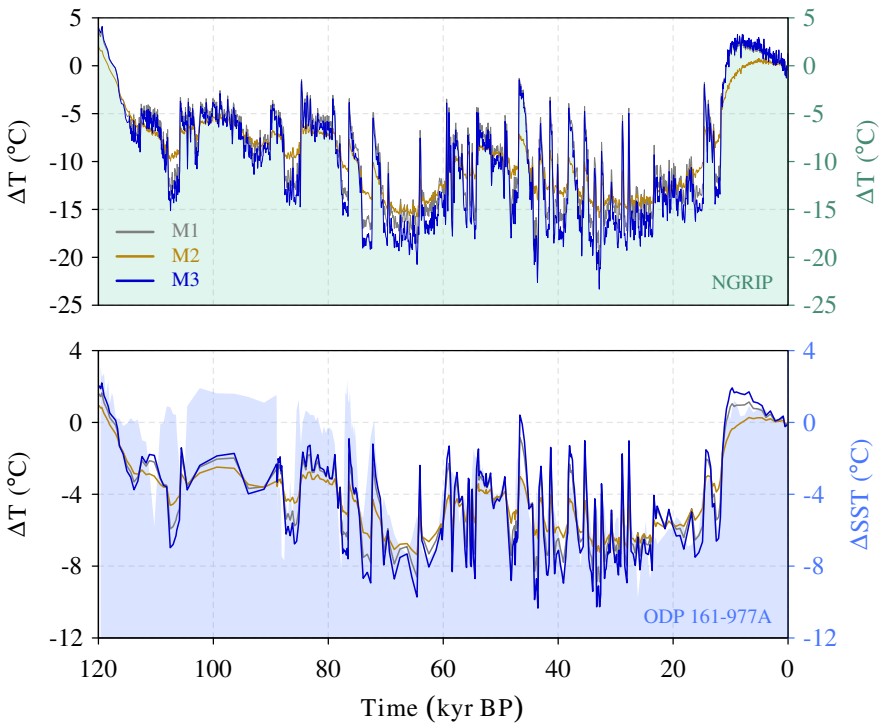

**Figure 3.** Temporal evolution of surface temperature anomalies (°C) relative to present day obtained in M1 (gray), M2 (gold) and M3 (blue) (solid curves) as compared to proxy-based temperature reconstructions (shaded areas) at: **a)** the NGRIP site, in Greenland (75.1°N, 42.32°W), (Vinther et al., 2009; Kindler et al., 2014) ; **b)** the ODP 161-977A location, in the eastern Alboran Sea (36.03°N, 1.95°W), (Martrat, 2007; Martrat et al., 2014).





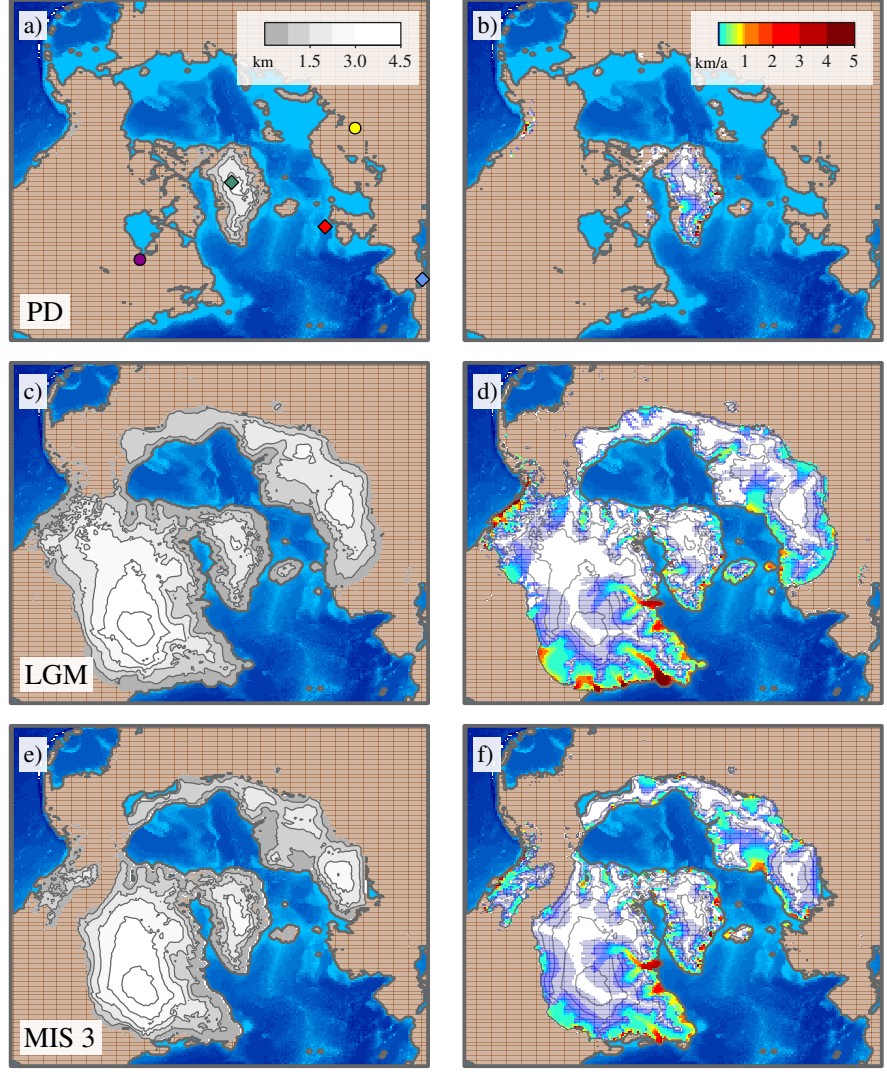

**Figure 4.** NH ice-sheet configurations at different stages of the last glacial-interglacial period as simulated under M3: **a)** present-day ice thickness (km) and **b)** present-day ice velocities (km a$^{-1}$). Panels **c)-d)** and **e)-f)** show the same information as **a)-b)** for the LGM and MIS3 stages, respectively. Colored diamonds (proxy-based reconstructions available) in panel **a)** show the locations of the NGRIP site (light green) and cores ODP 161-977A (light blue) and MD95-2006 (red), respectively. Colored dots (proxy-based reconstructions unavailable) show the locations of the two central sites considered at the LIS (purple) and the FIS (yellow).




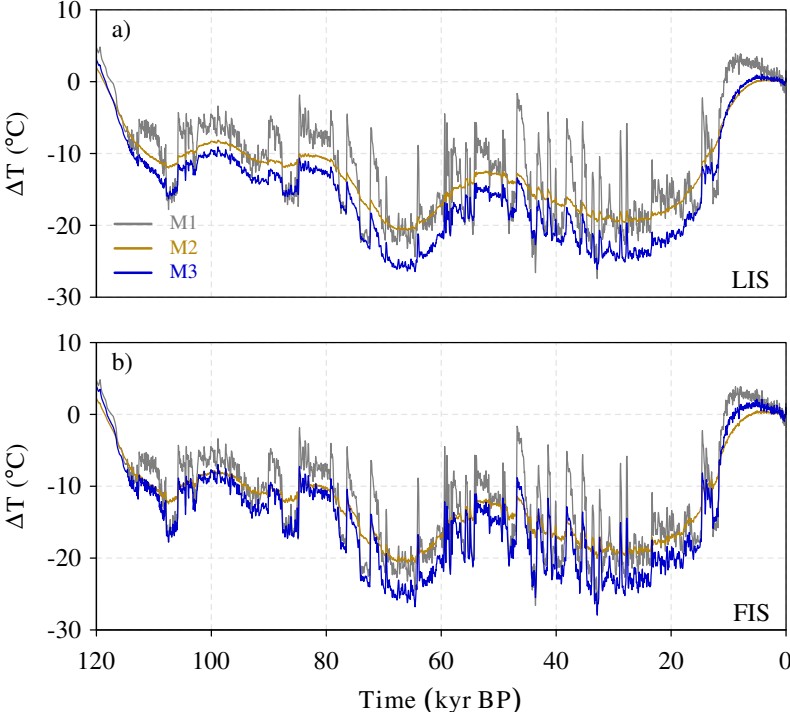

**Figure 5.** Temporal evolution of SAT anomalies (°C) relative to present day reconstructed under M1 (gray), M2 (gold) and M3 (blue) scenarios at two central locations of the LIS and the FIS in: **a)** North America and **b)** Eurasia.



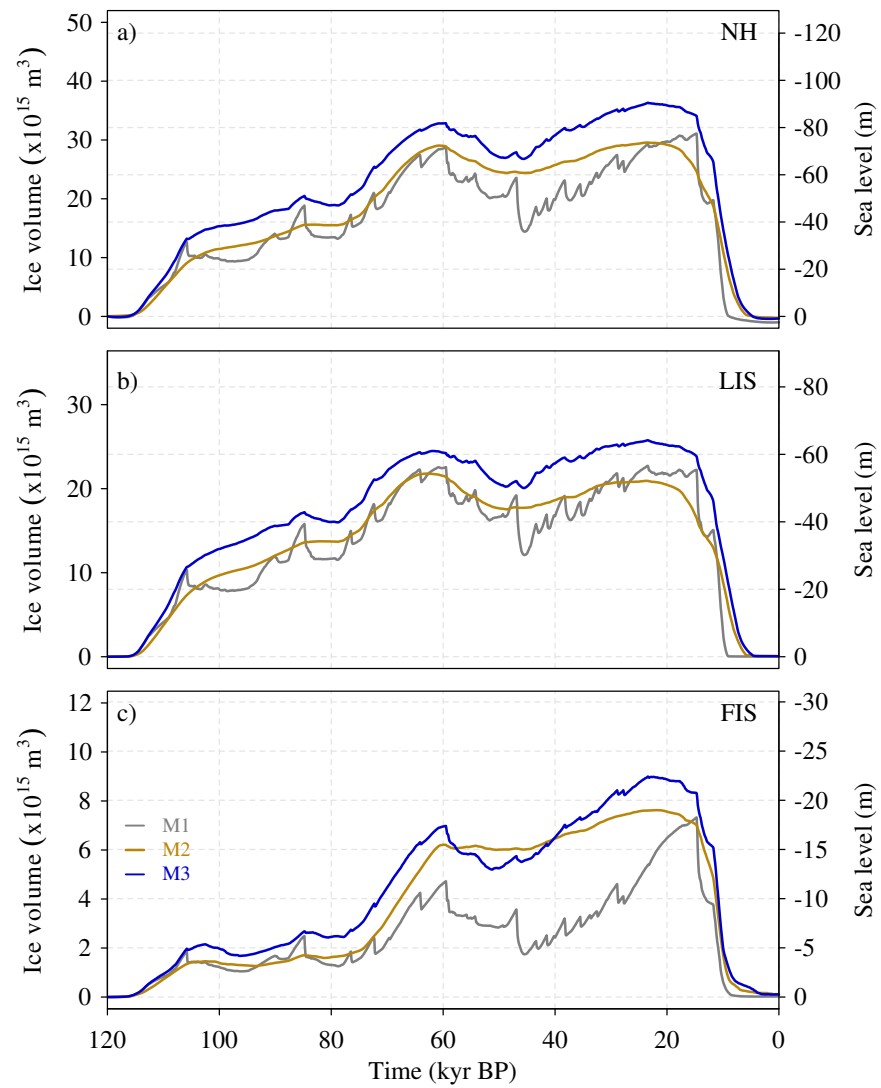

**Figure 6.** Temporal evolution of ice volume (m³) relative to initial conditions simulated in M1 (gray), M2 (gold) and M3 (blue) for: **a)** the NH domain; **b)** the LIS and **c)** the FIS. Ice volume variations have also been expressed in sea level equivalent units (m).