# Peer review of "A new approach for simulating the paleo evolution of the Northern Hemisphere ice sheets"

_Geoscientific Model Development, 2017_

## Referee Comment (RC1) · Anonymous Referee #1 · 22 Aug 2017

**Review of Banderas et al, 2017, GMDD**
* * *
This study proposes and tests a new approach for creating a transient climate forcing from a small set of equilibrium climate model simulations and a proxy record index (e.g. Greenland ice core data). In the traditional "index" method only two simulated climate states are used, essentially describing all temporal changes with one climate anomaly pattern. In the here newly proposed method, long-term variability ("orbital") is separated from short-term variability ("millennial"), with each connected to a different simulated climate pattern/anomaly. This modification improves one of the weakest features of the "index" method, namely that it assumes that all climate variability over the assessed period is spatially constant (as they all follow one pattern).
The new approach is certainly interesting and could be published in GMD. However, the manuscript lacks some important discussion, and is in places not very clear. I therefore suggest a large number of general and more specific comments that I think should be discussed before publication in GMD.

**GENERAL COMMENTS**

1. The novelty of this new approach is not made very clear in the manuscript. Adding an extra spatial pattern for a certain frequency range will largely improve the "index" method. Please emphasize this. Also, the explanation of the traditional method (M1) is not very clear (e.g. Page 2, line 32 to page 3 line 3). Please rewrite.

2. These methods ("index" and updated "index") assume that temperature variations reconstructed over Greenland are representative for the entire Northern Hemisphere. Is this valid? Please discuss.

3. Even though having an offline climate forcing can be very useful, lately efforts have been made in coupling climate models of intermediate complexity (even CLIMBER specifically) to ice sheet models (see for example Bauer and Ganopolski, 2017). The impact of the lack of climate-ice sheet feedbacks in your approach needs to be discussed.

4. What is the resolution of the CLIMBER-3a model, and how does this (low) resolution impact the (40x40km) ice-sheet model results?

5. Which factors are used in the PDD scheme? And are they kept constant over the domain and different simulations? Bauer and Ganopolski (2017) suggest that when using fixed PDD factors one cannot realistically simulate the glacial evolution of the Northern Hemisphere ice sheets. Please discuss.

6. The study will largely improve when more paleo data is included. For example, the Eurasian data set covering 25-10ka of Hughes et al. (2016) could be used to evaluate which method captures the FIS transition from glacial to Holocene best. Also, is there any data indicating a different behaviour of the LIS vs FIS over this

period (see page 8, lines 23-31)? And are you sure there is no other independent (proxy) temperature data for the last 120ka?

**SPECIFIC AND TECHNICAL COMMENTS**

The following list of suggestions is intended to improve the readability of the text:

Page 1, lines 4-5: Rewrite to: "The impact of the climatologies on the paleo evolution of the NG ice sheets is evaluated."

Page 1, line 5: change usual approach to "index approach"

Page 1, lines 5-7: please rephrase. Maybe first mention the climate anomaly field and ice-core index, and then add this to PD climate?

Page 1, line 9: "corrected to provide a perfect agreement", this is called tuning.

Page 1, line 10: "recent" is confusing, because it could mean recently measured, or recently published.

Page 1, line 11: change "usual" to "index"

Page 1, line 13: rephrase to "results in a too small ice volume"

Page 1, line 18: change to "variability and improves the transient ..."

Page 1, line 21: change to "need to be invoked to explain millennial-scale ice volume fluctuations."

Abstract and Discussion: would be helpful for the reader if the approaches/methods are numbered as in the rest of the text. For example: Our new method (M3)...

Page 2, line 2: move references to end of sentence, as these papers use proxy data.

Page 2, lines 9-10: These LIG values are not precise, but estimates. Add "roughly", "approximately" or similar.

Page 2, lines20-25; another paper that should be cited here is: Goelzer et al, 2016. Also, another approach to simulate paleo ice sheet evolution is by using ice sheet models with reduced complexity, forced by simple climate forcing (e.g. Langebroek et al., 2009).

Page 2, line 30: change to "lack of continuous spatially well distributed proxy data"

Page 2, line 31: "even based on simple assumptions" is very vague. Maybe leave out?

Page 2 line 32-page 3, line 3: please rephrase. Remember also that climatology is not the same as temperature.

Page 3, line 5: "latter" is not clear, do you mean "index" approach?

Page 3, line 7: change to "orbital climatic variations, while it is scaled following the characteristic time evolution of the index, which includes orbital and millennial-scale climate variability"

Page 3, line 9: change "two modes" to "orbital and millennial"

Page 3, lines 10-11: "As a result, this method can be expected to lead to a misrepresentation of millennial scale climate variability…"

Page 4, lines 12-13: Basal melting is no "surface" boundary condition. Please rewrite.

Page 4, line 16: What do you mean with linear atmospheric profile? Do you just adjust the temperatures with a temperature lapse rate depending on elevation? What is the lapse rate value (degC/km)?

Page 4, line 19: which PDD factors are used?

Page 4, line 20: Is basal melting not also depending on the ice thickness?

Page 4, line 21: change to "in regions where the ocean floor is below 450m …"

Page 4, line 25: definition of PD should be stated first time present-day is used

Page 4, lines 25-26: Change to: "PD climatology obtained from observational data, with simulated climate snapshots of the last glacial cycle and a time dependent index derived from proxy records."

Page 4, line 34: "using the ICE-5G topography"

Page 5, line 1: How do the ocean temperatures impact the ice sheet model results, if the basal melting is fixed?

Page 5, line 5: "index" instead of "usual"

Page 5, equations: Maybe I miss something, but wouldn't it be easier to use something like $T(t) = T0 + gamma(t) * dTorb$; with gamma = 1 for LGM, and gamma = 0 for PD ?

Page 5, line 11: delete "previous"

Page 5, line 11: preindustrial or rather PD?

Page 5, lines 13-14: delete "time" before index. "Thus, the index dictates the timing of both orbital and millennial-scale variability."

Page 5, equation (5): Maybe you have to emphasise that "gamma = alpha + beta"

Page 6, lines 22-23: Rewrite to: "… NGRIP ice-core location. This tuning to the NGRIP KV reconstruction also introduces a scaling of the synthetic…"

Page 7, line 12: Change "indirect measurements" to "reconstructions"

Page 7, line 13: Change " As an initial proof of consistency" to "We first compare"

Page 7, line 26: Change "climatologies" to "temperatures"

Page 7, line 32: change to: "… suggesting an too low amplitude …"

Page 8, line 1: It is very difficult to see the amplitude of the M2007 reconstruction. Maybe it would help if this is also shown as a (dashed?) line? What is the sample resolution of this core? Is it high enough to capture the high variability of the simulated temperature evolution?

Page 8, lines 15-16: Several geological time periods are mentioned here (Eemian, Holocene, MIS2 and MIS4). Their ages need to be stated. It would also be very helpful if these periods are indicated in the figures.

Page 8, lines 20-22: Change to: "… only at the millennial scale set by the difference between the PD and interstadial temperature fields used in…"

Page 8, line 24: change "reflecting the fact" into "meaning"

Page 8, line 33: "forcings to" instead of "methods in"

Page 8, line 34: Figures S2 and S3 do not really show any orbital climate variability

Page 9, line 5: change to: "… reconstruction, the climates of M2 and M2 are identical at orbital timescales, and only differ at …"

Page 9, line 5-7: Unclear, please rephrase. Do you mean that the orbital patterns are used to explain the millennial changes, and because the orbital changes are large, the response is also (too) large?

Page 9. Line 11: change " its larger orbital amplitude" to "tuning to the lower NGRIP temperature"

Page 9, 12-13: change to: "The temperature fluctuations in M3 incorporate both the larger orbital and the smaller millennial amplitude fluctuations compared to M1."

Page 9, line 15: SLE difference is maybe on average 20 m, but not "constant". Also it is not clear from the figures that this difference is larger for LIS than FIS. Please quantify by taking some mean.

Page 9, lines 15-16: "The intermediate case M2 follows more closely M1 in the LIS, but M3 in the FIS." What does this mean? Please elaborate.

Page 9, line 16: There is no clear drop at 55 ka, in Figure 6. Maybe it is rather around 58 ka, or 48ka? Please update, and indicate in Fig 6 which drop is meant.

Page 9, lines 22-28: What is the timing of these D/O events? Please indicate in Fig 5&6.

Page 9, line 23: Which positive feedbacks? Please discuss.

Page 9, line 33: I think Figure 5 should be cited here instead of Fig. 4

Page 9, line 34: "our view" is very vague. And there is not much data to support it. Maybe best to rewrite to say that M3 is the most advanced method or so?

Page 10, line 3: Change to "Figure 4b"

Page 10, line 5: the wording of "divides" is confusing here. Do you mean ice sheet divides or continental water divides?

Page 10, lines 4-7: The LGM distribution from M3 is very different from M1 and M2 (as shown as Supp figures). Maybe this could be more quantitatively compared to a dataset (Peltier?), and used as argumentation that M3 is the best method?

Page 10, line 11: add "Hughes et al., 2016"

Page 10, line 14: The Supplements do not really show climate variability.

Page 10, line 16: This is actually "a new method", not "2 methods".
Page 10, lines 22-23: Change to: "Depending on the frequency either the glacial-interglacial climate anomaly field (orbital variability) or the stadial-interstadial field (millennial) is varied."

Page 10, line 24: change to: "… and millennial-scale variation are tuned to fit the Greenland ice-core record."

Page 10, line 32: change to "The different climatologies have a large impact on the development of NH ice sheets…"

Page 11, line 9: change "these sites" to "this region".

Page 11, line 21: Change "Improving its representation" to "Including millennial-scale patterns"

Page 11, line 31: Hughes et al. (2016) suggests ~23m. Again, it would be helpful if the values are also indicated in the figures, as well as the timing of the LGM.

Page 11, line 34: Would be useful to add the sea-level curve from Fig. 1a in Fig. 6 in order to see the difference in reconstructed and simulated variability.

Page 12, line 1-2: Please also discuss here the missing feedbacks between climate and ice sheet in this offline method (e.g. albedo effect).

Page 12, line 13: Change "therefore" to "apply that to"

Figure 1: a) The sea-level curve is not used as forcing, or? Then please delete "forcing"

Figure 1: Is the VK index only derived from NGRIP? If not, please rewrite figure caption.

Figure 1: The shading is difficult to see, could you make it less transparent?

Figure 2: Can you add the locations of the analysed sediment cores?

Figure 3: "obtained by"; is it Martrat et al., 2014 or 2004?

Figure 4 is not mentioned in the text until after Fig 5&6. Maybe change the order?

Figure 6: What are the initial conditions, how much ice, and where? Maybe easier to make this graph relative to today? (is probably very similar)

Please make sure that the website storing the results is available.

**REFERENCES**

Bauer, E. and Ganopolski, A.: Comparison of surface mass balance of ice sheets simulated by positive-degree-day method and energy balance approach, Clim. Past, 13, 819-832, https://doi.org/10.5194/cp-13-819-2017, 2017.

Goelzer, H., Huybrechts, P., Loutre, M.-F., and Fichefet, T.: Last Interglacial climate and sea-level evolution from a coupled ice sheet–climate model, Clim. Past, 12, 2195-2213, https://doi.org/10.5194/cp-12-2195-2016, 2016.

Hughes, Anna L C; Gyllencreutz, Richard; Lohne, Øystein S; Mangerud, Jan; Svendsen, John Inge (2015): The last Eurasian ice sheets - a chronological database and time-slice reconstruction, DATED-1. *Boreas*, 45(1), 1-45.

Langebroek, P. M., Paul, A., and Schulz, M.: Antarctic ice-sheet response to atmospheric $CO_2$ and insolation in the Middle Miocene, Clim. Past, 5, 633-646, https://doi.org/10.5194/cp-5-633-2009, 2009.

---

## Referee Comment (RC2) · Anonymous Referee #2 · 26 Nov 2017

Banderas et al. study provides a method based on climate index to simulate the evolution of Northern Hemisphere ice sheet reconstructions through the past glacial period (110k - 10k). Instead of using one index, derived from NGRIP ice core record, or insolation changes, as traditionally done by many other previous works using the index approach to simulate NH glaciations, they provide three different indices, varying in the complexity, the most complex one including some millennial scale variability. They then use the 3D ice-sheet model GRISLI to simulate the transient evolution of the Laurentide and the Eurasian ice sheets through the last glacial period. They conclude that the index including the millennial scale variability provides the most satisfying results in terms of extent and volume of the two ice sheets at the LGM and during MIS3.

While the authors present the work as a novel approach, this is, to my opinion, only a

different way of using the index method and this is not particularly novel in the sense that previous studies were able to also get satisfying results with more simple index approach. In particular, if the aim of this study is to show the impact of including the millennial scale within the simulations, then I find that the result analysis is not enough to support the conclusions, especially from statistical and point of view. For example, the simulated LGM ice sheet extent underestimates the reconstructed Laurentide extent and overestimates the Eurasian one. In addition, I wonder why the authors did not derived the entire glacial period until 10k, since from LGM to 10k, the new DATED-1 reconstruction of Eurasian ice sheet extent could also provide a very strong way of validating the present work. Only few simulations with specific ice-sheet model parameters settings are presented here, whereas most of the ice-sheet parameters that have been chosen strongly impact on the final conclusions. To substantially strengthen this study, similar ice sheet simulations varying the key PDD parameters, calving and oceanic melt rates are necessary. In addition, you never mention the hydrology that you use at the base of the ice sheet. In GRISLI this particularly important because according to the criterion used, you can or not trigger the SSA on larger domains. This also could enhance the sensitivity to climate forcing and then show different response than the one you show here. From this point of view, assumptions on which this work relies are mentioned but poorly discussed or missing. Furthermore, the authors try to account for the millennial scale variability in their simulations but they removed the contribution from ocean by imposing 0 melting in the shelf expansion areas and keeping this fixed for the entire simulations. I find that this is a weakness of this approach and should be combined with additional simulations in which the imposed value could be also derived from index (2012), as done by Pollard and De Conto (2012) for example. Based on the results then the authors could strengthen their discussion and conclusions. I detail below my specific comments. In its current state, this study requires further substantial investigations and improvements before publications.

**General comments:**

**Validation of the simulations:** the validation work is not enough and requires more elaborations.

1- Why not running the experiment until 10k, this does not cost much more in terms of resources because you run at 40k and one simulation takes at max 12h-18h and you can actually also validate your results with DATED-1 (Hughes et al., 2016) for Eurasia. Because you mentioned many times in the manuscript that you lack of proxies for a proper validation. This is one way of doing it. I would like then to see your deglaciation simulations with the different indices and the match with DATED-1 extent or for example Patton et al. (2016) modelling work.

2- Could plot the extent of ICE-5G for the Laurentide and of DATED-1 at LGM on your Figure 4? So one can appreciate the performance of your index?

3- You can also validate the elevation changes over Greenland, since you have the NGRIP records etc.., which you never show. You could follow the paper by Quiquet et al. You do not use a lot the work by Kleman et al. (2013). They tried to bound the extent with proxies. So I advice you to you their reconstruction to support your simulations.

4- I am not convinced by the comparison between indices and SSTs is uncertain. Because you mentioned it several times, I would suggest to remove those parts. You don't know if air T° and SST always co-vary with a similar amplitude. To me, this part is a bit weak.

[Figure]

5- You never show how precipitations evolves your indices. Actually d18O of NGRIP is more representative of precipitation than temperatures. Please, also show precipitation.You could use speleothems from the Mediterranean and other places to validate your derived precipitations.

6- the index method main weakness is to not account for changes in circulation, therefore the climate-ice-sheet feedbacks are mainly missing. This is the most important weakness of this method. Please also mention it in the method and in the discussion.

**Methods:** I find the methods not clear enough

1- about the explanations of climate snapshots: please provides more informations about the resolution of the forcing, which matters a lot for downscaling. Provide also informations about your downscaling procedure.

2- I would like to see a Table of the main parameters used in the ice sheet simulations: in particular, PDD parameters, calving threshold, lapse rate, basal drag etc. . .All those parameters matters a lot in your case and you never discuss this.

3- NGRIP is not representative of North America and Eurasian circulation changes. This is one of the assumption you never discussed in the paper. In your Figure 5, for example, Eurasian and North America have the same trend. Which is not the case in proxies and strongly depends on the region. Why not comparing the transient simulated evolution of the CLIMBER experiments over Greenland for sure, but also over Eurasia and North America to see the difference generated by circulation and regional changes. Then insert a justification on the fact that you use Greenland ice core records to derived your indices. From this point of view, I don't see the improvement compared with traditional index methods. A way that would have been

perhaps more robust would have been to derived indices from the transient climate simulation itself over several regions, used several indices in your ice sheet simulations and add the part of missing variability to each index as you do with your method M3.

4- You never mention the criterion used to trigger the SSA on continents. GRISLI uses the Shelfy-stream approximation, this has a lot of importance for the simulated velocities and volume. A part about this aspect and discussing your choice is necessary to support your analysis.

5- I find highly necessary that you test your hypothesis against different values of oceanic melt rates and calving values because this also has a lot of influence on the transient evolution. When you do steady-state, it is important to prescribe reasonable value to affect the grounding line in a realistic way. This is even more important in the case of transient simulations. I would thus suggest to add new simulations, in addition to the ones already performed and presented here, varying those values and also test the importance of the calving criterion. Here you set ad-hoc at 0m/yr and larger at depth greater than 450 m. First of all, you really force the answer of your simulations by doing so and you artificially get rid of the calving issue. Second, 0m/yr might be valid for the LGM, but not for the other previous periods. Thus, You should test this assumption and change your method/value for oceanic forcing.

6- You chose to use the PDD method to simulate ablation. Why not using ITM for example, that would also provide a different view of the impact of your index and would be also perhaps more indicated to catch the millennial scale variability embedded in your indices. PDD might strongly dampened the effect of your indices here. In addition to this, you also use a fractionning of precipitation and snow that is very drastic, based on the temperature threshold of 2°C. (Why not using the method from Marsiat (1994) as many of GRISLI studies do, just to test a different criterion of precip/snow

fractionning?). One cannot test everything, but here you are looking at evolution of mass balance, so those aspects matter.

---

## Author Comment (AC1) · 17 Jan 2018

**A new approach for simulating the paleo evolution of the Northern Hemisphere ice sheets**

R. Banderas • J. Alvarez-Solas • A. Robinson • M. Montoya

**Referee #1**

This study proposes and tests a new approach for creating a transient climate forcing from a small set of equilibrium climate model simulations and a proxy record index (e.g. Greenland ice core data). In the traditional "index" method only two simulated climate states are used, essentially describing all temporal changes with one climate anomaly pattern. In the here newly proposed method, long-term variability ("orbital") is separated from short-term variability ("millennial"), with each connected to a different simulated climate pattern/anomaly. This modification improves one of the weakest features of the "index" method, namely that it assumes that all climate variability over the assessed period is spatially constant (as they all follow one pattern). The new approach is certainly interesting and could be published in GMD. However, the manuscript lacks some important discussion, and is in places not very clear. I therefore suggest a large number of general and more specific comments that I think should be discussed before publication in GMD.

**GENERAL COMMENTS**

1. The novelty of this new approach is not made very clear in the manuscript. Adding an extra spatial pattern for a certain frequency range will largely improve the "index" method. Please emphasize this. Also, the explanation of the traditional method (M1) is not very clear (e.g. Page 2, line 32 to page 3 line 3). Please rewrite.

We recognise that the novelty of the approach might not have come across clearly given some of the comments of the reviewers. To emphasise it and thus the relevance of our study we have modified the abstract to stress the problems of usual index methods and how we try to improve these:

"Offline forcing methods for ice sheet models often make use of an index approach in which temperature anomalies relative to present are calculated by combining a simulated glacial-interglacial climatic anomaly field, interpolated through an index derived from the Greenland ice-core temperature reconstruction, with present-day climatologies. An important drawback of this approach is that it clearly misrepresents climate variability at millennial timescales. The reason for this is that the spatial glacial-interglacial anomaly field used is associated with orbital climatic variations, while it is scaled following the characteristic time evolution of the index, which includes orbital and millennial-scale climate variability. The spatial patterns of orbital and millennial variability are clearly not the same, as indicated by a

wealth of models and data. As a result, this method can be expected to lead to a misrepresentation of climate variability and thus of the past evolution of Northern Hemisphere (NH) ice sheets. Here we illustrate the problems derived from this approach, and propose a new offline climate forcing method that attempts to better represent  the characteristic pattern of millennial-scale climate variability by including an additional spatial anomaly field associated with this timescale. "

We have also rephrased the explanation of the traditional method M1 in the introduction following  the reviewer's specific suggestion #3 below to make it more clear:

"Often an index approach is followed in which temperature anomalies relative to present are calculated by combining a simulated glacial-interglacial climatic anomaly field, interpolated through an index derived from the Greenland ice-core temperature reconstruction, with present-day climatologies."

2. These methods ("index" and updated "index") assume that temperature variations reconstructed over Greenland are representative for the entire Northern Hemisphere. Is this valid? Please discuss.

Both methods indeed assume that temperature variability reconstructed over Greenland is representative of the entire Northern Hemisphere (NH),  but this does not mean that either the amplitude or the sign was the same in the whole NH. That is, actually, the case in usual methods but not in our new method, which is one of the reasons why it represents an improvement. The reason is the millennial scale anomaly pattern  introduces its own (spatial) scaling. The details of this spatial pattern will depend on the particular climate model used to produce the climate anomaly fields, and might well improve with higher complexity and resolution.  This is included now in the Conclusions and Discussion section:

"Note that offline index methods assume that the temperature variability reconstructed over Greenland is representative of the entire Northern Hemisphere NH,  but this does not mean either that the amplitude or the sign is the same in the whole NH. This is, actually, the case in usual methods but not in our new method, which is one of the reasons why it represents an improvement. The reason is that the millennial scale anomaly pattern introduces its own (spatial) scaling. The details of this spatial pattern will depend on the particular climate model used to produce the climate anomaly fields, and might well improve with higher complexity and resolution. Most models agree in showing that NH temperature changes coeval with Greenland in response to northward heat transport changes caused by AMOC variations, the prevailing paradigm to explain glacial abrupt climate changes (e.g. Stouffer et al. 2006), and this is supported by comprehensive review of spatial coverage (Voelker et al. 2002), but this is not an assumption of our new index method."

3. Even though having an offline climate forcing can be very useful, lately efforts have been made in coupling climate models of intermediate complexity (even CLIMBER specifically) to ice sheet models (see for example Bauer and Ganopolski, 2017). The impact of the lack of climate-ice sheet feedbacks in your approach needs to be discussed.

We have included a caveat in the Conclusions and Discussion section to recognise the potential limitations of offline-forcing methods such as ours due to the lack of climate-ice sheet feedbacks. Nevertheless it should be beared in mind that our approach is meant precisely to improve offline methods when these are required (see also specific comment #72):

"Although our ice-sheet model accounts for the surface elevation change feedback on temperature and precipitation, other important climate-ice sheet feedbacks such as surface albedo changes are not represented. Note, however our goal is precisely to improve offline forcing methods, for which most of these feedbacks are inherently absent. It would nevertheless be interesting to investigate this issue further by coupling our ice-sheet model to a regional energy-moisture balance model where feedbacks such as the ice-albedo feedback, the effect of continentality and the orographic effect on precipitation are better represented."

4. What is the resolution of the CLIMBER-3a model, and how does this (low) resolution impact the (40x40km) ice-sheet model results?

The resolution of the atmospheric component of the CLIMBER-3$\alpha$ model is 7.5º × 22.5º (latitude × longitude). This low resolution is obviously a limitation of the example we give in our study, but the only manner in which the long time scales involved here can be investigated. Using a higher climate resolution model should provide both a more accurate representation of millennial-scale glacial climate variability and a more realistic forcing for the ice-sheet model, but because our conclusions depend only on the differences between the orbital and millennial scale temperature patterns we expect our results to be robust. Of course it would be interesting to assess this question further with more comprehensive and higher resolution models. We have included this briefly in the Conclusions and Discussion section:

"The climate model used to build the present-day, LGM, and interstadial fields used in this study is an intermediate complexity model with low spatial (latitude × longitude) resolution (7.5º × 22.5º). Using a more comprehensive and/or higher resolution model should provide both a more accurate representation of millennial-scale glacial climate variability and a more realistic forcing for the ice-sheet model. Nevertheless, we do not expect this to change our main conclusions. To the extent that orbital and millennial-scale anomaly fields are different, our new forcing method should provide a better representation of the climate of the LGP. We expect this result to be robust against the use of different climate models."

5. Which factors are used in the PDD scheme? And are they kept constant over the domain and different simulations? Bauer and Ganopolski (2017) suggest that when using fixed PDD factors one cannot realistically simulate the glacial evolution of the Northern Hemisphere ice sheets. Please discuss.

All PDD parameters are kept constant in all simulations over the entire domain. Daily temperature is computed from summer and annual mean temperatures, with its standard deviation $\sigma$ set to 5 ºC. Ablation is calculated using melting rates for snow and ice ($C_{snow}$ and $C_{ice}$) of 0.003 and 0.008 mwe/(PDD). As mentioned by Bauer and Ganopolski (2017), using fix PDD factors it is not possible to realistically simulate the glacial evolution of the NH ice sheets in coupled climate - ice-sheet models. The reason being that the increase of $CO_2$ and insolation after the LGM is not efficient enough to satisfactorily simulate the deglaciation when using a PDD approach. Here, and for all the index methods, the deglaciation is explicitly driven by an imposed increase in temperatures, thus the mentioned problem does not appear. Nevertheless, our goal is not to provide the most realistic simulation, which should include coupling with the climate system, higher resolution and a better representation of surface mass balance processes, but rather to highlight a deficiency and to improve current offline methods. To recognise this we have included the former discussion on this issue in the Experimental Setup section, together with a table containing all relevant ice-sheet model parameters:

"All PDD parameters are kept constant in all simulations over the entire domain (see Table 1 for the exact parameter values). Note that as indicated by Bauer and Ganopolski (2017), using fix PDD factors it is not possible to realistically simulate the glacial evolution of the NH ice sheets in coupled climate - ice-sheet models. The reason being that the increase of $CO_2$ and insolation after the LGM is not efficient enough to satisfactorily simulate the deglaciation when using a PDD approach. Here, and for all the index methods, the deglaciation is explicitly driven by an imposed increase in temperatures, thus the mentioned problem does not appear. Nevertheless, our goal is not to provide the most realistic simulation, which should include coupling with the climate system, higher resolution, and a better representation of surface mass balance processes, but rather to highlight and overcome an important deficiency of current offline methods."

6. The study will largely improve when more paleo data is included. For example, the Eurasian data set covering 25-10ka of Hughes et al. (2016) could be used to evaluate which method captures the FIS transition from glacial to Holocene best. Also, is there any data indicating a different behaviour of the LIS vs FIS over this period (see page 8, lines 23-31)? And are you sure there is no other independent (proxy) temperature data for the last 120 ka?

We have now increased the comparison with data in several ways. In Figure 3, we have included δ¹⁸O (‰ SMOW) variations from northern European Alps (47.38ºN, 10.15ºE) stalagmites, considered as a proxy for surface air temperatures in central Europe (Moseley et al. 2014) as well as δ¹⁸O (‰ VPDB) variations inferred from Fort Stanton Cave (33.3ºN, 105.3ºW) speleothems (Asmerom et al. 2010) as a proxy for precipitation in southwestern North America (see reviewer 2's comment #5). In Figure 4, the extent of ICE-5G (Peltier, 2004) and DATED-1 (Hughes et al. 2016) have been added to the LGM ice extent simulated under M3 (see also reviewer 2's comment 2). In Figure, 6 available LGM ice volume estimates for the LIS and the EIS (Denton and Hughes, 1981; Clark and Mix, 2002; Tarasov et al. 2012, Clark and Tarasov, 2014, Hughes et al. 2016) as well as the ice volume evolution for the deglaciation from DATED-1 (Hughes et al. 2016) have also been included (see comment #60 related to this below and comment #1 by reviewer 2). The comparison with these proxy data has been discussed in different points of the new version of the manuscript:

In section 3.1:

[revised manuscript text omitted]

**SPECIFIC AND TECHNICAL COMMENTS**

The following list of suggestions is intended to improve the readability of the text:

We appreciate the reviewer's detailed list of specific suggestions. To facilitate the review we number these:

ABSTRACT

1. Page 1, lines 4-5: Rewrite to: "The impact of the climatologies on the paleo evolution of the NH ice sheets is evaluated." Done.

2. Page 1, line 5: change usual approach to "index approach" Done. However, in order to distinguish the usual or traditional approach that has been used up to now from our new approach we have chosen to rephrase this as "the usual index approach".

3. Page 1, lines 5-7: please rephrase. Maybe first mention the climate anomaly field and ice-core index, and then add this to PD climate? We have rephrased this as "Often an index approach is followed in which temperature anomalies relative to present are calculated by combining a simulated glacial-interglacial climatic anomaly field, interpolated through an index derived from the Greenland ice-core temperature reconstruction, with present-day climatologies." (see also major comment 1 above).

4. Page 1, line 9: "corrected to provide a perfect agreement", this is called tuning. We have replaced "corrected" by "tuned".

5. Page 1, line 10: "recent" is confusing, because it could mean recently measured, or recently published. We have replaced this by "recently published".

6. Page 1, line 11: change "usual" to "index" As mentioned above, we think it is important to highlight that this is the approach that has been generally used up to now; thus we have rephrased this again as "usual index". Note all the methods described involve indices, the difference is the new method makes use of additional (millennial-scale) climate anomalies.

7. Page 1, line 13: rephrase to "results in a too small ice volume" The phrase is correct as it is in the current version of the manuscript.

8. Page 1, line 18: change to "variability and improves the transient ..." Done.

9. Page 1, line 21: change to "need to be invoked to explain millennial-scale ice volume fluctuations." Done.

10. Abstract and Discussion: would be helpful for the reader if the approaches/methods are numbered as in the rest of the text. For example: Our new method (M3)... Although we have generally followed this approach, we prefer not to do so in the Abstract and Discussion sections, which in our opinion should be read smoothly without reference to particular technical details such as the names of the experiments.

SECTION 1

11. Page 2, line 2: move references to end of sentence, as these papers use proxy data. Done.

12. Page 2, lines 9-10: These LIG values are not precise, but estimates. Add "roughly", "approximately" or similar. Done.

13. Page 2, lines 20-25; another paper that should be cited here is: Goelzer et al, 2016. Also, another approach to simulate paleo ice sheet evolution is by using ice sheet models with reduced complexity, forced by simple climate forcing (e.g. Langebroek et al., 2009). The references to Goelzer et al. (2016) and Langebroek et al. (2009) have been included here.

14. Page 2, line 30: change to "lack of continuous spatially well distributed proxy data" Done.

15. Page 2, line 31: "even based on simple assumptions" is very vague. Maybe leave out? Done.

16. Page 2 line 32-page 3, line 3: please rephrase. Remember also that climatology is not the same as temperature. This has been rephrased as "The time-varying temperature climatology".

17. Page 3, line 5: "latter" is not clear, do you mean "index" approach? Indeed. To make this more clear "the latter" has been replaced by "this index approach".

18. Page 3, line 7: change to "orbital climatic variations, while it is scaled following the characteristic time evolution of the index, which includes orbital and millennial-scale climate variability". Done.

19. Page 3, line 9: change "two modes" to "orbital and millennial". Done.

20. Page 3, lines 10-11: "As a result, this method can be expected to lead to a misrepresentation of millennial scale climate variability..." Done.

21. Page 4, lines 12-13: Basal melting is no "surface" boundary condition. Please rewrite. This has been rephrased as "Boundary conditions include the surface mass balance (SMB) and basal melting".

2. METHODOLOGY. 2.1. Model description

22. Page 4, line 16: What do you mean with linear atmospheric profile? Do you just adjust the temperatures with a temperature lapse rate depending on elevation? What is the lapse rate value (degC/km)? This is right. The linear atmospheric profile results from adjusting temperatures with a temperature lapse rate which depends on elevation. In this study, we consider different values for the annual (8 ºC/km) and the summer (6.5 ºC/km) lapse rates. To make this more clear we have rephrased it as "GRISLI accounts for changes in elevation at each time step considering a linear atmospheric vertical profile for temperature with different lapse rates in summer and in the annual mean to account for the smaller summer atmospheric vertical stability (Table 1)".

23. Page 4, line 19: which PDD factors are used? This issue is related to the main comment 5 above. The values of the PDD factors are now included in the manuscript together with all relevant ice-sheet model parameters (Table 1).

24. Page 4, line 20: Is basal melting not also depending on the ice thickness? No it does not; note here basal melting in floating parts is a constant background value chosen in order to reproduce reasonable glacial background conditions and to focus only on the atmospheric variations.

25. Page 4, line 21: change to "in regions where the ocean floor is below 450m ..." We think the current phrasing is better; nevertheless to clarify this we have rephrased "depth" as "the ocean depth".

SECTION 2: METHODOLOGY, 2.1: MODEL DESCRIPTION

26. Page 4, line 25: definition of PD should be stated first time present-day is used
We think it is more appropriate to define PD here because it refers not generally to present-day but to the terminology used for the experimental setup.

27. Page 4, lines 25-26: Change to: "PD climatology obtained from observational data, with simulated climate snapshots of the last glacial cycle and a time dependent index derived from proxy records." Done.

28. Page 4, line 34: "using the ICE-5G topography" Done.

29. Page 5, line 1: How do the ocean temperatures impact the ice sheet model results, if the basal melting is fixed? The oceanic effect is prescribed in this work; the aim of this paper is not to study the effect of varying ocean temperatures but atmospheric ones only, in order to isolate the effect of our new approach. Actually, the effect of varying oceanic temperatures is assessed in a separate paper, currently in open discussion in Climate of the Past Discussions. Nevertheless, following reviewer 2's suggestion #5 on Methods we have performed additional simulations to consider different background basal melting values and added the following discussion in the Experimental setup:

"Increasing background basal melting values modulates the response of NH ice sheets to millennial-scale forcing (see Supplementary Material). A more detailed analysis of the effect of oceanic changes on NH ice sheets will be addressed in future work."

30. Page 5, line 5: "index" instead of "usual" As above, we prefer to write "usual index approach"

31. Page 5, equations: Maybe I miss something, but wouldn't it be easier to use something like T(t) = T0 + gamma(t) * dTorb; with gamma = 1 for LGM, and gamma = 0 for PD ?, The equation that the referee is proposing is equivalent to the one used in our study. Using one or the other simply depends on whether the present or the LGM climate is recovered for $\gamma$=1, and different studies use different approaches. For example, Zweck and Huybrechts (2005) used the formulation suggested by the reviewer (a "glacial index" that is 1 for the LGM and 0 for present), while Charbit et al. (2007) used the same formulation as we did. This is arbitrary and has no impact in the results. Nevertheless to make it clear we explicitly mention it in the manuscript: "Note that the $\gamma$ index can be defined as here (Charbit et al. 2007) or instead as a glacial index (1-$\gamma$) that is 0 for the present and 1 for the LGM (e.g. Marshall et al. 2000; 2002; Zweck and Huybrechts 2005)."

32. Page 5, line 11: delete "previous" Done.

33. Page 5, line 11: preindustrial or rather PD? This is actually preindustrial; present-day is actually used in the whole text in this sense. We have explained this now in the Introduction the first time that "present" appears: "(note the present is meant here and after to indicate preindustrial conditions)".

34. Page 5, lines 13-14: delete "time" before index. "Thus, the index dictates the timing of both orbital and millennial-scale variability." Done.

35. Page 5, equation (5): Maybe you have to emphasise that "gamma = alpha + beta"
    This has been explicitly included now in section 2.2.2: "Thus $\gamma = \alpha + \beta$"

**2.2.3. METHOD 3**

36. Page 6, lines 22-23: Rewrite to: "... NGRIP ice-core location. This tuning to the NGRIP KV reconstruction also introduces a scaling of the synthetic..." Done.

37. Page 7, line 12: Change "indirect measurements" to "reconstructions" Done.

38. Page 7, line 13: Change " As an initial proof of consistency" to "We first compare" Done.

39. Page 7, line 26: Change "climatologies" to "temperatures" Done.

40. Page 7, line 32: change to: "... suggesting an too low amplitude ..." The expression we have used is correct as it is in the manuscript.

41. Page 8, line 1: It is very difficult to see the amplitude of the M2007 reconstruction. What is the sample resolution of this core? Is it high enough to capture the high variability of the simulated temperature evolution? Following reviewer 2's comments (see #4 and #5) we have suppressed the comparison with Mediterranean SSTs and replaced it by a comparison with a proxy for precipitation. Nevertheless, the new Figure 3 has been made taking into account reviewer 1's suggestion to make it more clear.

42. Page 8, lines 15-16: Several geological time periods are mentioned here (Eemian, Holocene, MIS2 and MIS4). Their ages need to be stated. It would also be very helpful if these periods are indicated in the figures. We have inserted the approximate dates for the Eemian and the Holocene and rephrased this sentence as follows: "the temperature variations obtained by all methods in both sites show warmer climate conditions at the Eemian (ca. 125 ka BP) with respect to the Holocene (10 ka BP to present day) and colder temperatures throughout the LGP.

43. Page 8, lines 20-22: Change to: "... only at the millennial scale set by the difference between the PD and interstadial temperature fields used in..." We prefer to keep our current formulation.

44. Page 8, line 24: change "reflecting the fact" into "meaning". We prefer to keep our current formulation.

**3.2 RECONSTRUCTION OF NH ICE SHEETS**

45. Page 8, line 33: "forcings to" instead of "methods in" Done.

46. Page 8, line 34: Figures S2 and S3 do not really show any orbital climate variability
We have corrected this to include only Figure 6.

47. Page 9, line 5: change to: "... reconstruction, the climates of M2 and M2 are identical
at orbital timescales, and only differ at ..." Done.

48. Page 9, line 5-7: Unclear, please rephrase. Do you mean that the orbital patterns are
used to explain the millennial changes, and because the orbital changes are large, the
response is also (too) large? Indeed, this is precisely the problem with the "usual
index" method. To make this more clear we have rephrased this as follows: "Indeed,
the orbital anomalies used by standard index methods to represent millennial changes
are larger than the millennial-scale anomalies. Thus the forcing and the response are
overestimated".

49. Page 9. Line 11: change " its larger orbital amplitude" to "tuning to the lower NGRIP
temperature" Done.

50. Page 9, 12-13: change to: "The temperature fluctuations in M3 incorporate both the
larger orbital and the smaller millennial amplitude fluctuations compared to M1."
Done.

51. Page 9, line 15: SLE difference is maybe on average 20 m, but not "constant". Also it
is not clear from the figures that this difference is larger for LIS than FIS. Please
quantify by taking some mean. The difference is not obvious because the scales are
different, but Figure 6c shows for the FIS the differences are always below 10 m; by
contrast Figure 6b shows for the LIS they can reach up to 20 m. We have nevertheless
rephrased this as follows: "[SLE differences...] are generally larger for the LIS than
for this FIS".

52. Page 9, lines 15-16: "The intermediate case M2 follows more closely M1 in the LIS,
but M3 in the FIS." What does this mean? Please elaborate. It means that the
evolution of the LIS in M2 resembles that in M1, but the evolution of the FIS in M2
rather resembles that in M3. We have rephrased this in this way now: "Regarding the
evolution of the LIS, M2 resembles M1 more than M3, but for the evolution of the
FIS, M2 resembles more M3 than M1".

53. Page 9, line 16: There is no clear drop at 55 ka, in Figure 6. Maybe it is rather around
58 ka, or 48ka? Please update, and indicate in Fig 6 which drop is meant. The referee
is right. We were referring to the drop at 48 ka.

54. Page 9, lines 22-28: What is the timing of these D/O events? Please indicate in Fig 5&6. We think this is not really necessary once we have corrected the error in the date as above.

55. Page 9, line 23: Which positive feedbacks? Please discuss. This has been rephrased as "the positive feedbacks between surface elevation and temperature as well as precipitation".

56. Page 9, line 33: I think Figure 5 should be cited here instead of Fig. 4 Done.

57. Page 9, line 34: "our view" is very vague. And there is not much data to support it. Maybe best to rewrite to say that M3 is the most advanced method or so? We have decided to suppress "our view". Actually with the new comparisons included it is clear now that this is the case.

58. Page 10, line 3: Change to "Figure 4b" Done.

59. Page 10, line 5: the wording of "divides" is confusing here. Do you mean ice sheet divides or continental water divides? We have replaced the word "divides" by "ice sheets".

60. Page 10, lines 4-7: The LGM distribution from M3 is very different from M1 and M2 (as shown as Supp figures). Maybe this could be more quantitatively compared to a dataset (Peltier?), and used as argumentation that M3 is the best method?

As also suggested by reviewer #2 (validation comment 2), we have now included the extent of ICE-5G (Peltier, 2004) and DATED-1 (Hughes et al. 2016) in Figure 4 as well as in Figures S1 and S2 (See major comment 6 above for a more detailed explanation).

61. Page 10, line 11: add "Hughes et al., 2016" Done.

62. Page 10, line 14: The Supplements do not really show climate variability. We agree and have suppressed the reference to the supplementary material here.

63. Page 10, line 16: This is actually "a new method", not "2 methods". The reviewer is right; we have corrected this as suggested.

64. Page 10, lines 22-23: Change to: "Depending on the frequency either the glacial-interglacial climate anomaly field (orbital variability) or the stadial-interstadial field (millennial) is varied." Done.

65. Page 10, line 24: change to: "... and millennial-scale variation are tuned to fit the Greenland ice-core record." Done.

66. Page 10, line 32: change to "The different climatologies have a large impact on the development of NH ice sheets..." Done.

67. Page 11, line 9: change "these sites" to "this region". Done.

68. Page 11, line 21: Change "Improving its representation" to "Including millennial-scale patterns" Done.

69. Page 11, line 31: Hughes et al. (2016) suggests ~23m. Again, it would be helpful if the values are also indicated in the figures, as well as the timing of the LGM. Done. Actually following comment #1 of reviewer 2 we have also included the ice volume evolution for the EIS during the deglaciation in Figure 6 and the following discussion in section 3.2:

"As a consequence, of all three methods only M3 agrees with the available LGM minus present SLE reconstructions within their ranges of uncertainties, both for the LIS and the FIS."

In addition, we mention the 23 m estimate in the Conclusions and Discussion.

70. Page 11, line 34: Would be useful to add the sea-level curve from Fig. 1a in Fig. 6 in order to see the difference in reconstructed and simulated variability. We have explicitly avoided direct comparison with the reconstructed sea-level curve because these generally provide inferences of global sea-level changes, which complicates the evaluation of our simulated NH ice volume timeseries against the paleorecord. This has been explained in the text and it is the reason to show the comparison with the available reconstructions for specific time slices:

"Although sea-level records provide essential information to interpret past ice-volume variations, continuous highly-resolved sea-level reconstructions are scarce and frequently rely on an insufficient temporal control. In addition, they generally provide inferences of global sea-level changes. This complicates the evaluation of our simulated NH ice volume timeseries against the paleorecord. However, the contribution to sea level of individual ice sheets can be assessed at specific time slices such as the LGM, for which reconstructions are indeed available."

71. Page 12, line 1-2: Please also discuss here the missing feedbacks between climate and ice sheet in this offline method (e.g. albedo effect). We have included here a brief

discussion to recognise the lack of coupling to the climate system and the missing feedbacks:

"In a similar manner, although our ice-sheet model accounts for the surface elevation change feedback on temperature and precipitation, other important climate-ice sheet feedbacks such as surface albedo changes are not represented. Note, however our goal is precisely to improve offline forcing methods, for which most of these feedbacks are inherently absent. It would nevertheless be interesting to investigate this issue further by coupling our ice-sheet model to a regional energy-moisture balance model where feedbacks such as the ice-albedo feedback, the effect of continentality and the orographic effect on precipitation are better represented."

72. Page 12, line 13: Change "therefore" to "apply that to" Done.

73. Figure 1: a) The sea-level curve is not used as forcing, or? Then please delete "forcing". The sea-level curve is actually used as forcing as indicated in section 1.

74. Figure 1: Is the VK index only derived from NGRIP? If not, please rewrite figure caption. The KV reconstruction is obtained by combining Vinther et al. (2009) for the Holocene and Kindler et al. (2014) for the last glacial period. As explained in the manuscript, Vinther et al. (2009) provides a Holocene temperature reconstruction over Greenland inferred from six different ice core locations. The reconstruction by Kindler et al. (2014) applies exclusively to the NGRIP site and covers the whole LGP. Caption in Figure 1 has been rephrased in order to prevent any confusion:

"The black curve shows the evolution of temperature anomalies (°C) relative to present over Greenland from which the index is derived (Vinther et al. 2009; Kindler et al. 2014)."

75. Figure 1: The shading is difficult to see, could you make it less transparent? Done.

76. Figure 2: Can you add the locations of the analysed sediment cores? This is already included in Figure 4.

77. Figure 3: "obtained by"; is it Martrat et al., 2014 or 2004?

We have removed the comparison with SST reconstructions following reviewer #2's suggestions (see comments 4 and 5 below). In the new version of the manuscript we have included central Europe and southwestern North American proxies for air temperature and precipitation.

78. Figure 4 is not mentioned in the text until after Fig 5&6. Maybe change the order? Figure 4 is already mentioned in page 8, line 28, before Figures 5 and 6 are mentioned.

79. Figure 6: What are the initial conditions, how much ice, and where? Maybe easier to make this graph relative to today? (is probably very similar) Initial conditions at 120 ka BP are actually simply present-day conditions; this was stated in the Experimental Setup. Nevertheless, to make this more clear we have rephrased the sentence where we refer to them in the Experimental setup as "Initial topographic conditions are provided by present surface and bedrock elevations built from the ETOPO1 dataset (Amante and Eakins 2009) and ice thickness (Bamber et al. 2001).

80. Please make sure that the website storing the results is available. Supplementary data for this manuscript can be found in this link: http://www.palma-ucm.es/data/ism-forcing/

**Referee #2**

Banderas et al. study provides a method based on climate index to simulate the evolution of Northern Hemisphere ice sheet reconstructions through the past glacial period (110k - 10k). Instead of using one index, derived from NGRIP ice core record, or insolation changes, as traditionally done by many other previous works using the index approach to simulate NH glaciations, they provide three different indices, varying in the complexity, the most complex one including some millennial scale variability. They then use the 3D ice-sheet model GRISLI to simulate the transient evolution of the Laurentide and the Eurasian ice sheets through the last glacial period. They conclude that the index including the millennial scale variability provides the most satisfying results in terms of extent and volume of the two ice sheets at the LGM and during MIS3. While the authors present the work as a novel approach, this is, to my opinion, only a different way of using the index method and this is not particularly novel in the sense that previous studies were able to also get satisfying results with more simple index approach. In particular, if the aim of this study is to show the impact of including the millennial scale within the simulations, then I find that the result analysis is not enough to support the conclusions, especially from statistical and point of view.

For example, the simulated LGM ice sheet extent underestimates the reconstructed Laurentide extent and overestimates the Eurasian one. In addition, I wonder why the authors did not derived the entire glacial period until 10k, since from LGM to 10k, the new DATED-1 reconstruction of Eurasian ice sheet extent could also provide a very strong way of validating the present work. Only few simulations with specific ice-sheet model parameters settings are presented here, whereas most of the ice-sheet parameters that have been chosen strongly impact on the final conclusions. To substantially strengthen this study, similar ice

sheet simulations varying the key PDD parameters, calving and oceanic melt rates are necessary.

In addition, you never mention the hydrology that you use at the base of the ice sheet. In GRISLI this particularly important because according to the criterion used, you can or not trigger the SSA on larger domains. This also could enhance the sensitivity to climate forcing and then show different response than the one you show here. From this point of view, assumptions on which this work relies are mentioned but poorly discussed or missing. Furthermore, the authors try to account for the millennial scale variability in their simulations but they removed the contribution from ocean by imposing 0 melting in the shelf expansion areas and keeping this fixed for the entire simulations. I find that this is a weakness of this approach and should be combined with additional simulations in which the imposed value could be also derived from index (2012), as done by Pollard and De Conto (2012) for example. Based on the results then the authors could strengthen their discussion and conclusions. I detail below my specific comments. In its current state, this study requires further substantial investigations and improvements before publications.

**General comments**

**Validation of the simulations**: the validation work is not enough and requires more elaborations.

Before responding to specific comments, we would like to clarify that our main goal is not to validate the realism of the ice-sheet simulations, but to offer a new index methodology and to show its advantages. We use GRISLI to show that including a stadial-interstadial field weighted by a new spectrally-derived index has relevant implications for the interpretation of the ice-sheet behavior on millennial time scales. This conclusion is independent of the ice sheet model used. We have, nevertheless, carried out some additional simulations as requested to show that the main message of this paper does not rely on the particularities of the parameters chosen here (see point #5 of the methods).

1- Why not running the experiment until 10k, this does not cost much more in terms of resources because you run at 40k and one simulation takes at max 12h-18h and you can actually also validate your results with DATED-1 (Hughes et al., 2016) for Eurasia. Because you mentioned many times in the manuscript that you lack of proxies for a proper validation. This is one way of doing it. I would like then to see your deglaciation simulations with the different indices and the match with DATED-1 extent or for example Patton et al. (2016) modelling work.

The meaning of this comment is difficult to interpret. In fact, our simulations run up to present day, as our figures clearly show, not at 40k. Thus we do not understand what the reviewer means. Concerning validation, please see the comment above. This work does not aim to provide the most realistic ice-sheet model simulation possible. Nevertheless, we have

added the volume evolution of the Eurasian ice sheet during the deglaciation provided by DATED-1 for comparison in current figure 6.

2- Could plot the extent of ICE-5G for the Laurentide and of DATED-1 at LGM on your Figure 4? So one can appreciate the performance of your index?

We have followed the referee's suggestion and added a comparison to DATED-1 LGM extension. Concerning the Laurentide, we followed the referee's suggestion and added the ice distribution of ICE-5G.

In section 3.2:

"In terms of the extent of NH ice sheets at the LGM, M3 appears to be the best of the three methods, showing the most satisfactory agreement with reconstructions: ICE-5G (Peltier, 2004) for the LIS and DATED-1 (Hughes et al. 2016) for the FIS (Figure 4c; see also the Supplementary Material). Major deficiencies are found in the southeastern margin of the Scandinavian Ice Sheet (SIS), the southwestern border of the LIS and the northern part of the Cordilleran Ice Sheet (CIS), where the ice extent is underestimated as compared to reconstructions and northwestern Siberia, where it is overestimated. In M1 and M2, these discrepancies with reconstructions are more evident. Furthermore, in the corridor that separates the CIS and the LIS a significant ice retreat is observed that is absent in M3 (see Supplementary Material)."

3- You can also validate the elevation changes over Greenland, since you have the NGRIP records etc.., which you never show. You could follow the paper by Quiquet et al. You do not use a lot the work by Kleman et al. (2013). They tried to bound the extent with proxies. So I advice you to you their reconstruction to support your simulations.

We understand the reviewer's concerns about validation. For this reason we have included in the new version of the manuscript the extent of ICE-5G (Peltier, 2004) and DATED-1 (Hughes et al. 2016) as well as a set of proxy data to validate the phasing and timing of our new method (see comment #5 below for a more detailed information).

4- I am not convinced by the comparison between indices and SSTs is uncertain. Because you mentioned it several times, I would suggest to remove those parts. You don't know if air T and SST always co-vary with a similar amplitude. To me, this part is a bit weak.

We understand the reviewer's concerns. Thus we have removed this part as well as the old Figure 3b (see comment below) and S1. In the new version of the manuscript we have considered speleothems from Fort Stanton Cave (33.3ºN, 105.3ºW; southwestern North America) as a proxy for precipitation as well as from northern European Alps (47.38ºN,

10.15ºE) as a proxy for surface temperature. This is discussed in section 3.1 (see also comment #5 below and major comment 6 by reviewer 1):

"We further evaluate the three methods through comparison with available temperature and precipitation reconstructions derived from speleothems in Central Europe (the Alps) and North America. Time series of SAT in central Europe show an overall qualitative agreement among all three methods (Figure 3b), which reproduce the phasing and timing of millennial-scale climate variability registered  in terrestrial records from the northern European Alps (Moseley et al. 2014). Nevertheless, there are important quantitative differences among the three methods, with M3 showing the SAT changes with the largest amplitudes, followed by M1, and M2 the smallest ones."

5- You never show how precipitations evolves your indices. Actually d18O of NGRIP is more representative of precipitation than temperatures. Please, also show precipitation.You could use speleothems from the Mediterranean and other places to validate your derived precipitations.

We have followed the referee's suggestion replacing the old Figure 3c with a new panel showing $\delta^{18}O$ (‰ VPDB) variations from Fort Stanton Cave (33.3ºN, 105.3ºW; southwestern North America) as a proxy for precipitation together with our simulated precipitation from M1, M2 and M3. Discussion from this new Figure 3c is now included in the new version of the manuscript (section 3.1; see also major comment 6 by reviewer 1):

"Furthermore, the simulated temporal evolution of precipitation in southwestern North America reveals important differences among the two methods. In particular, M1 shows an antiphase relationship with respect to simulated precipitation in M3 on millennial time scales (Figure 3c). The reason for this lies in the differences that exist within the spatial patterns of orbital and millennial scale climate variability in this particular region. While the millennial-scale pattern shows slightly wetter conditions during the stadial (i.e. colder climate) as compared to the interstadial ($\delta P_{mil}<1$) in southwestern North America, the orbital spatial pattern exhibits slightly drier conditions at the LGM (i.e. colder climate) as compared to PD conditions ($\delta P_{orb}<1$). Available proxy information indicates that increased precipitation in this area is associated with NH cooling (Asmerom et al. 2010) as opposed to the pervasive NH signal inferred from a wealth of records (Wang et al. 2001; NGRIP members, 2004) which evidences that wetter conditions generally occur during interstadials. Thus, M3 successfully reproduces precipitation variability as interpreted by proxies in this particular region, a result that cannot be achieved by means of the usual index approach."

In addition, the Discussion and Conclusions section includes the following:

"The time series derived from these methods are compared at several locations with the available proxy data: the Greenland ice-core record and a reconstruction of temperature and

precipitation based on $\delta^{18}O$ variations in speleothems located in southwestern North America and central Europe, respectively. By construction, the new method provides a perfect agreement with the ice-core record, improving the performance of previous methods. For temperature the three methods follow a similar evolution, as dictated by the Greenland ice-core record, but the new method shows a larger amplitude. For precipitation, the new method yields a very different time evolution as a result of the spatial millennial-scale anomaly pattern which successfully reproduces the phasing and timing of $\delta^{18}O$ variability in southwestern North America on millennial time scales, a result that cannot be achieved by the old method."

6- The index method main weakness is to not account for changes in circulation, therefore the climate-ice-sheet feedbacks are mainly missing. This is the most important weakness of this method. Please also mention it in the method and in the discussion.

Indeed offline methods do not account explicitly for changes in circulation, however a nuance has to be pointed out here: both methods (the classical index and our improved one) assume that temperature variability reconstructed over Greenland is representative of the entire Northern Hemisphere (NH). In the classical one (which includes the anomalies of LGM-present day as the only spatial field) this usually translates in having a similar amplitude and the same sign all over the NH even at millennial time scales. By including a new spatial pattern (reflecting stadial-interstadial anomalies) that now handles the millennial-scale evolution, this is not necessarily the case. A stadial to interstadial transition (even in the form of a single snapshot) indirectly captures the effects that the changes in circulation have on temperature and precipitation (mainly centered on the North Atlantic). Therefore the mentioned changes in circulation are now indirectly considered by means of a linearized and weighted signal that can be more or less complex depending on the climate model used.

In the example we give in the paper, CLIMBER-3α simulates a stadial-to-interstadial warming that is of the same sign for the whole NH but not zonally homogeneous. This is the reason why the millennial anomalies shown in the Laurentide are half the magnitude of those shown in Eurasia (please see M3 in Figure 5), an effect that cannot be captured by construction with the classic method. In other climate models a stadial-to-interstadial anomaly might, for example, imply a cooling in the North Pacific. Therefore if that signal is used under our proposed methodology that would imply including an anomaly of opposite sign in that region when simulating the response of the ice sheets at millennial scale. The details of this spatial pattern will depend on the particular climate model used to produce the climate anomaly fields, and might well improve with higher complexity and resolution. Again, this phenomenon would not be captured under the usual approach. This is the main reason why our methodology represents an improvement when comparing different offline methods. We agree with the referee in the sense that the main weakness of any offline method is the absence of a direct consideration of the climate-ice-sheet feedbacks. But precisely

because of this, our methodology represents an improvement. Nevertheless, a caveat on feedbacks have now been added to the manuscript in the Conclusions and Discussion section (see also specific comment #72 by reviewer 1):

"Although our ice-sheet model accounts for the surface elevation change feedback on temperature and precipitation, other important climate-ice sheet feedbacks such as surface albedo changes are not represented. Note, however our goal is precisely to improve offline forcing methods, for which most of these feedbacks are inherently absent. It would nevertheless be interesting to investigate this issue further by coupling our ice-sheet model to a regional energy-moisture balance model where feedbacks such as the ice-albedo feedback, the effect of continentality and the orographic effect on precipitation are better represented."

**Methods:** I find the methods not clear enough

1- About the explanations of climate snapshots: please provides more informations about the resolution of the forcing, which matters a lot for downscaling. Provide also informations about your downscaling procedure.

We realize this was missing in the previous version and are grateful for this suggestion. We now include the following text in section 2.2 (The forcing methods):

"Due to the relatively low resolution of the atmospheric model (7.5° x 22.5°; latitude × longitude), we perform a two-step interpolation procedure to obtain the forcing fields at the resolution of the ice-sheet model. First, the fields were interpolated conservatively to the ice-sheet model grid. Then to eliminate artefacts related to model resolution, Gaussian smoothing (also conservative) was applied with a standard deviation of 250 km. Several smoothing windows were tested, with the final choice representing the minimum amount of smoothing necessary to ensure that sharp boundaries between the atmospheric grid cells could not be distinguished on the ice sheet model grid."

2- I would like to see a Table of the main parameters used in the ice sheet simulations: in particular, PDD parameters, calving threshold, lapse rate, basal drag etc. All those parameters matters a lot in your case and you never discuss this.

We agree it is useful to provide this information. We have added a table that is referenced at the end of section 2.1 as follows:

"A table of some key ice sheet model and climate forcing parameters has been included for reference (see Table 1)."

3- NGRIP is not representative of North America and Eurasian circulation changes. This is one of the assumption you never discussed in the paper. In your Figure 5, for example,

Eurasian and North America have the same trend. Which is not the case in proxies and strongly depends on the region. Why not comparing the transient simulated evolution of the CLIMBER experiments over Greenland for sure, but also over Eurasia and North America to see the difference generated by circulation and regional changes. Then insert a justification on the fact that you use Greenland ice core records to derived your indices. From this point of view, I don't see the improvement compared with traditional index methods. A way that would have been perhaps more robust would have been to derived indices from the transient climate simulation itself over several regions, used several indices in your ice sheet simulations and add the part of missing variability to each index as you do with your method M3.

Here two different points are raised which we address separately:

First, we did not carry a CLIMBER-3α transient simulation of the whole glacial. This is too expensive as it contains an oceanic GCM. Second (please see also the response to point #6), it is true that Eurasia and North America have a similar trend as shown by Figure 3. The orbital index controls the long-term evolution of the forcing by scaling two fields (glacial and present-day) which are primarily shaped by the effects of the ice sheets themselves on climate. However, on millennial time scales, the inclusion of a new index that weighs the stadial and interstadial fields no longer affects equally Eurasia and North America because the main temperature anomalies are now centered in the Nordic Seas (as suggested by data). This is the reason why M3 shows greater anomalies in Eurasia than in North America. The millennial scale anomaly pattern introduces its own (spatial) scaling. The details of this spatial pattern will depend on the particular climate model used to produce the climate anomaly fields, and might well improve with higher complexity and resolution. We think this a strong point of the paper and we should have stated it more clearly. We have now done so and added more information about it in the Conclusions and Discussion section:

"Note that offline index methods assume that the temperature variability reconstructed over Greenland is representative of the entire NH, but this does not mean either that the amplitude or the sign is the same in the whole NH. This is, actually, the case in usual methods but not in our new method, which is one of the reasons why it represents an improvement. The reason is the millennial scale anomaly pattern introduces its own (spatial) scaling. The details of this spatial pattern will depend on the particular climate model used to produce the climate anomaly fields, and might well improve with higher complexity and resolution. Most models agree in showing NH temperature changes in the NH coeval with Greenland in response to northward heat transport changes caused by AMOC variations, the prevailing paradigm to explain glacial abrupt climate changes (e.g. Stouffer et al. 2006) and that this is supported by comprehensive review of spatial coverage (Voelker et al. 2002), but this is not an assumption of our new index method."

4- You never mention the criterion used to trigger the SSA on continents. GRISLI uses the Shelfy-stream approximation, this has a lot of importance for the simulated velocities and volume. A part about this aspect and discussing your choice is necessary to support your analysis.

Again, we agree this was missing and necessary. We have now added additional information related to the triggering of SSA in section 2.1:

"In this model configuration, inland ice that is frozen to the bed is treated using SIA dynamics. When the base of the ice sheet becomes temperate (ie, there is water at the base), or when the ice is floating, then SSA dynamics apply. The basal friction is calculated as a linear function of the basal velocity that is proportional to effective pressure, $\boldsymbol{\tau}_b = C \cdot N_{eff} \cdot \mathbf{u}_b$."

While the treatment of the dynamics can affect the configuration of the simulated ice sheet, this is more likely to affect thickness (and thus ice volume) rather than spatial extent. To the first order, the latter is determined by the imposed climatic forcing. The simulations we performed are representative of a reasonable configuration of the ice sheet, and most importantly, the parameters are the same for all forcing methods. Therefore, the experiments allow for comparison between the forcing methods, which is the primary goal of the paper. Also, the current hybrid (SSA-SIA) configuration is shared with previous papers (Alvarez-Solas et al, 2011; Alvarez-Solas et al; 2013).

5- I find highly necessary that you test your hypothesis against different values of oceanic melt rates and calving values because this also has a lot of influence on the transient evolution. When you do steady-state, it is important to prescribe reasonable value to affect the grounding line in a realistic way. This is even more important in the case of transient simulations. I would thus suggest to add new simulations, in addition to the ones already performed and presented here, varying those values and also test the importance of the calving criterion. Here you set ad-hoc at 0m/yr and larger at depth greater than 450 m. First of all, you really force the answer of your simulations by doing so and you artificially get rid of the calving issue. Second, 0m/yr might be valid for the LGM, but not for the other previous periods. Thus, You should test this assumption and change your method/value for oceanic forcing.

The transient effect of the oceanic forcing on millennial-time scales is beyond the scope of this paper. This and related calving issues are now largely discussed in a new paper (Alvarez-Solas et al, 2017 CPD). However, we agree with the referee when pointing out that the chosen value of the oceanic melt rate largely affects the position of the grounding line. We have, therefore, studied this aspect by carrying out a new set of simulations where we explored a range of background values of the oceanic melt. This changes the position of the grounding line and thus the long-term simulated volume of the Eurasian ice sheet. The magnitude of the response of the ice sheet to millennial-scale forcing is accordingly modulated. This, however, does not affect the main conclusion of the paper in terms of the

advantages of our new forcing methodology. Nonetheless a new supplementary figure has now been added to show the response of the ice sheet to this new ensemble. This is discussed in the Experimental Setup section:

"Increasing background basal melting values modulates the response of NH ice sheets to millennial-scale forcing (see Supplementary Material). A more detailed analysis of the effect of oceanic changes on NH ice sheets will be addressed in future work".

6- You chose to use the PDD method to simulate ablation. Why not using ITM for example, that would also provide a different view of the impact of your index and would be also perhaps more indicated to catch the millennial scale variability embedded in your indices. PDD might strongly dampened the effect of your indices here. In addition to this, you also use a fractionning of precipitation and snow that is very drastic, based on the temperature threshold of 2∘C. (Why not using the method from Marsiat (1994) as many of GRISLI studies do, just to test a different criterion of precip/snow fractionning?). One cannot test everything, but here you are looking at evolution of mass balance, so those aspects matter.

Again, we do not aim at the most realistic ice-sheet simulation but to improve offline method. Our model might well have flaws but what we do here is provide an improvement of offline methods that we hope other modellers can take after. This is why this manuscript has been submitted to GMD. We recognise, however, that this message has not come fully across. We should have probably put a bit more weight on the possibilities that the new index itself provides to offline methodologies and index-forced ice sheet runs. For example, any PMIP3-like GCM now that accounts for stadial and interstadial-like runs can use our method to estimate the most likely evolution of the climate on millennial time scales during glacial period taking only a few snapshot. Furthermore that information can be used for ice-sheet modellers groups for forcing their models. We have tried to make this more clear in the new version of the manuscript by including the following in the abstract:

"Offline forcing methods for ice sheet models often make use of an index approach in which temperature anomalies relative to present are calculated by combining a simulated glacial-interglacial climatic anomaly field, interpolated through an index derived from the Greenland ice-core temperature reconstruction, with present-day climatologies. An important drawback of this approach is that it clearly misrepresents climate variability at millennial timescales. The reason for this is that the spatial glacial-interglacial anomaly field used is associated with orbital climatic variations, while it is scaled following the characteristic time evolution of the index, which includes orbital and millennial-scale climate variability. The spatial patterns of orbital and millennial variability are clearly not the same, as indicated by a wealth of models and data. As a result, this method can be expected to lead to a misrepresentation of climate variability and thus of the past evolution of NH ice sheets. Here we illustrate the problems derived from this approach, and propose a new offline climate forcing method  that attempts to better represent  the characteristic pattern of millennial-scale

climate variability by including an additional spatial anomaly field associated with this timescale. "